# Placental miR-340 mediates vulnerability to activity based anorexia in mice

Mariana Schroeder[1,2], Mira Jakovcevski [2], Tamar Polacheck[1,2], Yonat Drori[1,2], Alessia Luoni[2], Simone Röh[3], Jonas Zaugg[4,5], Shifra Ben-Dor [6], Christiane Albrecht [4,5] & Alon Chen [1,2]

Anorexia nervosa (AN) is a devastating eating disorder characterized by self-starvation that mainly affects women. Its etiology is unknown, which impedes successful treatment options leading to a limited chance of full recovery. Here, we show that gestation is a vulnerable window that can influence the predisposition to AN. By screening placental microRNA expression of naive and prenatally stressed (PNS) fetuses and assessing vulnerability to activity-based anorexia (ABA), we identify miR-340 as a sexually dimorphic regulator involved in prenatal programming of ABA. PNS caused gene-body hypermethylation of placental miR-340, which is associated with reduced miR-340 expression and increased protein levels of several target transcripts, GR, Cry2 and H3F3b. MiR-340 is linked to the expression of several nutrient transporters both in mice and human placentas. Using placenta-specific lentiviral transgenes and embryo transfer, we demonstrate the key role miR-340 plays in the mechanism involved in early life programming of ABA.

---

[1] Department of Neurobiology, Weizmann Institute of Science, 7610001 Rehovot, Israel. [2] Department of Stress Neurobiology and Neurogenetics, Max Planck Institute of Psychiatry, 80804 Munich, Germany. [3] Department of Translational Research in Psychiatry, Max Planck Institute of Psychiatry, 80804 Munich, Germany. [4] Institute of Biochemistry and Molecular Medicine, University of Bern, 3012 Bern Switzerland. [5] Swiss National Center of Competence in Research, NCCR TransCure, University of Bern, 3012 Bern, Switzerland. [6] Bioinformatics and Biological Computing Unit, Biological Services, Weizmann Institute of Science, 7610001 Rehovot Israel. Correspondence and requests for materials should be addressed to M.S. (email: mariana_schroeder@psych.mpg.de) or to A.C. (email: alon_chen@psych.mpg.de)

Eating disorders (ED) are destructive mental disorders that dramatically decrease quality of life and have life-threatening consequences. They are ten times more common in women than in men and represent one of the most common psychiatric disorders, affecting up to 4.5% of women at some point in their lives[1]. Anorexia nervosa (AN), the most severe ED, accounts for the highest mortality rate of all mental illnesses (about 10%)[2]. Despite being so immensely destructive to health and affecting such a significant number of young women, the understanding of AN remains poor due to its multi-factorial and complex origins. As a consequence, treatment is sparse, unspecific and unlikely to lead to full recovery[3].

Due to the significantly greater prevalence of AN in women, most of the clinical literature and preclinical models have aimed at providing a rationale for specific factors precipitating AN disproportionately in females at the time of puberty[4]. Susceptibility to EDs is often associated with early life stress or trauma[5,6]. However, it is important to make a distinction between the different types of "early life trauma" and how they may affect subsequent predisposition to disease. Trauma during childhood may serve as a trigger for an existing predisposition, while gestational or even pre-gestational trauma may be more involved in programming the predisposition or protecting from it, depending on the severity, timing, and extent of the insult[7]. As such, increased risk of bulimia nervosa and mixed eating disorders was reported among girls who were exposed to either prenatal stress (PNS) or early postnatal life stress. In contrast, this was not true for AN[8]. Yet, early postnatal and childhood trauma are both frequently referred to as "early life", but they are in fact taking place at different developmental windows, exerting potentially very different effects on the susceptibility for the different eating disorders in the offspring.

In any case, stress alone does not necessarily prompt EDs but rather a combination of an innate predisposition pooled with the additive effect of early life trauma is more likely to lead to disease[5,9]. This concept coincides with the developmental origin of health and disease hypothesis, and has received compelling epidemiological support from animal studies. Early life stress, particularly in the form of intrauterine maternal emotional stress or nutrition-related (high fat diet/deprivation), may ultimately predispose male and female offspring to different diseases[10–12]. The fetal perception of the intrauterine environment is mediated by the placenta, a feto-maternal organ that connects the developing fetus to the uterine wall to allow nutrient uptake, waste elimination, and gas exchange via the mother's blood supply. Although the placenta has long been considered a "sexless" organ, in fact given its predominantly fetal origin, it carries the same genetic information as the fetus and is therefore differentially sensitive to environmental stimuli[13] and fetal hormones[14]. The placenta of one sex may also possess a greater ability to respond and buffer against selective environmental insults[15], as is the case for the sex-specific placental contribution to nutritional programming in the offspring[16]. It is broadly recognized that compromised placental function can have both short- and long-lasting consequences for the developing fetus, e.g., with maternal hypoxia, pre-eclampsia, or placental insufficiency[17], which are closely related to a specific stage of development and fetal sex. Finally, distinct sex-dependent structural and functional placental factors have been repeatedly reported in normal pregnancies (reviewed in ref. [15]). This further suggests that the physiological and molecular basis for the sex-specific developmental trajectory toward susceptibility to disease may originate in utero, through placental disparities between the sexes despite exposure to a similar maternal environment.

Recently, microRNAs (miRs) have received major attention in the context of placenta-associated abnormalities[18]. Specific placental miRs were found to be associated with several pregnancy disorders and therefore are strong candidates for the mediation of gestational programming[19]. In the present study, we explored this possibility by comparing placental differences in miR expression between opposite sex siblings of undisturbed and PNS pregnancies in parallel to adolescent susceptibility to activity-based anorexia (ABA). The ABA model successfully mimics the choice of exercise above eating, relying on limited food intake while allowing unlimited access to running wheels (RW)[20]. While first modeled in rats, the ABA model has also been widely examined in mice[21] with varying criteria[22]. It is characterized by dramatic weight loss as a consequence of decreased food intake[21], hyperactivity and circadian disruption in the running pattern[23].

To test our hypothesis that susceptibility to AN may originate in utero, we examined placental miR expression in control and PNS mice and exposed a parallel cohort to the ABA protocol during adolescence. We demonstrate that susceptibility to ABA is associated with endogenous levels of placental miR-340 and overexpression dramatically increased the susceptibility to ABA. We propose that programming of ABA vulnerability originates in utero, and renders specifically female offspring vulnerable to metabolic challenges during adolescence.

## Results

**ABA affects mainly adolescent females and is abolished by PNS.** To examine the vulnerability of female mice to ABA, we exposed them to a modified version of the ABA protocol[24] when they reached adolescence. We allowed the animals to habituate to voluntary wheel running for 1 week with free access to food, followed by 5 days of free access to wheels combined with gradual food restriction (FR) in the form of 3–4 h of food access a day, during the dark cycle (Fig. 1a). An undisturbed control group with no wheels and free access to food as well as a further FR group with no access to wheels was added to control for the impact of FR on body weight (BW). The results revealed similar BW in all control female groups with free access to food. FR alone did not induce perdurable weight loss despite the reduction in food consumption, suggesting that the number of calories consumed was enough to maintain their BW if the necessary adjustments were made (such as reduction in general activity). In contrast, when exposed to the ABA protocol (FR + RW), the animals split into two sub-groups, denominated either "Resistant" (Res) or "Anorexic" (ABA). Overall, 40% of the control females developed ABA and 60% were resistant (Fig. 1b). Classification of the animals into each category was performed retrospectively by cluster analysis, based on the following parameters during the FR phase of the protocol: (1) BW loss; (2) Food intake in kcal; (3) Food intake recovery (% from the first FR day); (4) Circadian disruption (running distance in km during the light phase); (5) Total distance ran with FR (total activity) and (6) Time until collapse (in days). ABA females displayed dramatic weight loss compared to all other groups (Fig. 1c). Weight loss resulted from a gradual reduction in voluntary food intake (Fig. 1d). The FR group and Res females adapted and increased their kcal intake during the FR period, but ABA females did not (Fig. 1d). While Res females showed an intact circadian running pattern (Fig. 1e), ABA females showed disrupted running with high activity levels during the light phase (Fig. 1f) and running up to 12 km/day during the FR phase (Fig. 1g). In the long term (about 6 weeks after the end of the intervention), all females exposed to food restriction during adolescence weighed more than undisturbed controls, but did not differ according to predisposition to ABA (Fig. 1h). In order to better characterize each female mouse in this experiment, we assigned them an "ABA score" composed by the

sum of the $Z$ scores of the ABA parameters. Accordingly, ABA females (as revealed by the previous cluster analysis) scored significantly higher on this new scale than Res females (Fig. 1i). Finally, we examined gene expression in the hypothalamus, focusing on genes that have been specifically linked to anorexia in women and in various animal models as potential candidates. Hypothalamic dysregulation in anorexia has been reported for the melanocortin, the neurotrophic and the serotonin systems and the HPA axis. We chose representative genes of these systems that repeatedly emerged as potential mediators of the anorexic phenotype: Agouti-related peptide (AgRP)[25], Melanocortin receptor 4 (Mc4R)[26], Serotonin receptor (Htr)1a[27], Bdnf[28], Arginine vasopressin (Avp)[29] and Hypocretin/Orexin (Hcrt)[30]. We found that, as expected, AgRP, Htr1a, Avp and Hcrt were dysregulated in recovered ABA animals (Fig. 1j).

In order to test the potential mechanistic link between early life stress and later predisposition to ABA, we performed PNS on a further set of females starting 1 day after the detection of the copulation plug and until GD.16.5, that induced the expected stress-related responses in the mother and the fetuses (Supplementary Fig. 1A). The resulting offspring, including males, were then exposed to the ABA protocol upon reaching adolescence (Fig. 1k). We next calculated the $Z$ scores for each ABA parameter according to their strength of prediction as determined by a two-step cluster analysis in the original female group (Fig. 1l, raw data in Supplementary Fig. 2). We found that 30-day-old adolescent males and PNS offspring of both sexes scored similarly to Res females in intake recovery (Fig. 1m), days until collapse (Fig. 1n), BW change (Fig. 1o), circadian disruption (Fig. 1p), food intake (Fig. 1q), and total activity with FR (Fig. 1r). Thus, all males and PNS females were largely resistant to ABA, scoring similarly to Res control females in the ABA score (Fig. 1s). When examining basal metabolic parameters in this group, the PNS females appear to be programmed to become overweight, rather than underweight, and this may be at least one of the reasons underlying their resistance to ABA (Supplementary Fig. 1B).

**PNS placentas show downregulation of miR-340 by gene-body DNA methylation**. To elucidate the early origins of vulnerability/resistance to ABA, we next focused on gestational aspects and placental adaptations in control vs. PNS male and female offspring. BW gain in dams subjected to stress when pregnant (PNS dams) was significantly lower than in controls, despite similar food intake (Fig. 2a, b) and they also showed increased corticosterone levels (Fig. 2c). PNS did not affect fetal weight, but did reduce placental weight in female offspring only, suggesting an increase in placental efficiency in this group (Fig. 2d−f). Accordingly, placental 11βhsd$_2$, the enzyme that converts active corticosterone into inactive cortisone, was decreased and fetal corticosterone levels were increased exclusively in female PNS fetuses (Fig. 2g, h). This suggests that females were exposed to higher levels of endocrine stress signals compared to their male siblings, an effect reported previously[31].

To gain better insight into the mechanisms that mediate the differences between controls and PNS, we next compared the placental miRs expression profile using miR arrays. We identified several miRs differentially expressed in PNS placentas with miR-340 being most severely affected by the manipulation (Fig. 2i and Supplementary data 1). We then validated the array findings by qPCR and detected a clear sexually dimorphic expression pattern, since miR-340 was expressed in very low levels in control and PNS male placentas (Fig. 2j). Importantly, expression of miR-340, as shown by in situ hybridization, appears to be restricted to the junctional zone of the placenta in particular on the labyrinth side, a major site of feto-maternal exchange and hormone production respectively (Fig. 2k).

To elucidate the regulating mechanism of miR-340 by PNS, we performed a bioinformatic analysis on the Rnf130 gene, which hosts miR-340 in its second intron. Like miR-340, Rnf130 was downregulated by PNS (Supplementary Fig. 3A). Intronic miRs are frequently found in a sense orientation and thus are frequently transcribed together with their host gene[32]. The analysis revealed that Rnf130 has a CpG island about 800 bp long around its promoter region and the first exon (Fig. 2l). Furthermore, miR-340 was previously suggested to be regulated by DNA methylation in different types of cancer[33]. To examine this possibility, we used the DNA methyltransferase inhibitor 5-aza in an in vitro assay and showed that increasing concentrations of the inhibitor led to increasingly higher levels of miR-340, implying DNA methylation in its regulation (Fig. 2m). We next examined the methylation levels in the CpG Island of the Rnf130 promoter and on the miR-340 sequence through bisulfite conversion and pyrosequencing in control and PNS female placentas. While DNA methylation at promoters is widely accepted to be associated with transcriptional repression, recent studies suggested that gene body methylation also plays a critical role in gene regulation[34]. Interestingly, the CpG Island in the Rnf130 promoter region displayed very low levels of methylation, both in female control and PNS placentas (Supplementary Fig. 3B). In contrast, the two CpGs located on the sequence of miR-340 revealed high methylation in both groups, with CpG$_2$ significantly more methylated in PNS than in control female placentas (Fig. 2n), which also show global hypermethylation compared both to control females and males (Fig. 2o). Taken together, these findings suggest that placental miR-340 is a sexually dimorphic miRNA robustly affected by PNS and regulated through gene body DNA methylation.

**MiR-340 targets Nr3c1/GR, Cry2, and H3f3b in the placenta**. In order to explore the mechanism through which placental miR-340 exerts its effects on ABA predisposition, we next performed bioinformatic predictions on potential targets focusing on genes involved in placental functioning and structure. The conservation of the miR-340 seed matches on the 3′UTR of these targets are shown in Supplementary Fig. 4. Based on those predictions, we performed a Taqman custom gene expression array comparing placentas of control and PNS females. Among the potential targets, Sirt7, Nr3c1 (which encodes the GR), Hdac9, Histone 3 mammalian isoform, family 3b (H3f3b) and Ythdf3 were upregulated in placentas of PNS females (Supplementary Table 1). We expanded the analysis using RT-PCR, adding additional samples and performing correlations between the expression of miR-340 and the putative targets (Fig. 3a−f). GR (Fig. 3b), Hdac9 (Fig. 3c), Cry2 (Fig. 3d), and H3f3b (Fig. 3e) showed a significant inverse correlation with the expression of miR-340 in these samples, while Sirt7 (Fig. 3a) and Ythdf3 did not (Fig. 3f). When examining the PNS group alone, the strongest correlations were evident for GR/GR, Cry2/CRY2, and H3f3b/H3F3b (Supplementary Fig. 5a), which also showed increased placental protein levels (Fig. 3g−i). In contrast, Hdac9/HDAC9 did not show a similar effect (Supplementary Fig. 5b). To explore the link between the miR-340 and the relevant targets further, we next infected BeWo placental cells with a lentivirus designed to knockdown (KD) miR-340 and thus mimic the effects of PNS. As expected, the KD virus significantly reduced the expression of miR-340 in the cells (Fig. 3j). In addition, while mRNA levels did not change significantly in any of the putative targets (Fig. 3k), we found an increase in protein levels of GR, CRY2, and H3F3b (Fig. 3l), but again not in HDAC9 (Supplementary Fig. 5B).

Given the variability within the control female group in vulnerability to ABA, we next focused on this group in an

additional experiment where we analyzed the relative expression of miR-340 in 28 female samples from an additional six control pregnant females. Nine out of the 28 placentas (32%) showed miR-340 levels 25% above the average (Fig. 3m), which resembles the number of ABA-prone females in our first group (40%). Remarkably, the levels of miR-340 were affected by the sex of the

adjacent fetus. Females located between two males showed the lowest levels of miR-340 (Fig. 3n), suggesting prenatal androgen exposure may be linked to miR-340 expression levels and the subsequent resistance to ABA. Furthermore, we found an inverse correlation with the expression of miR-340 for *GR*, *Cry2*, and *H3f3b* in this group (Fig. 3o−q). Finally, we examined the

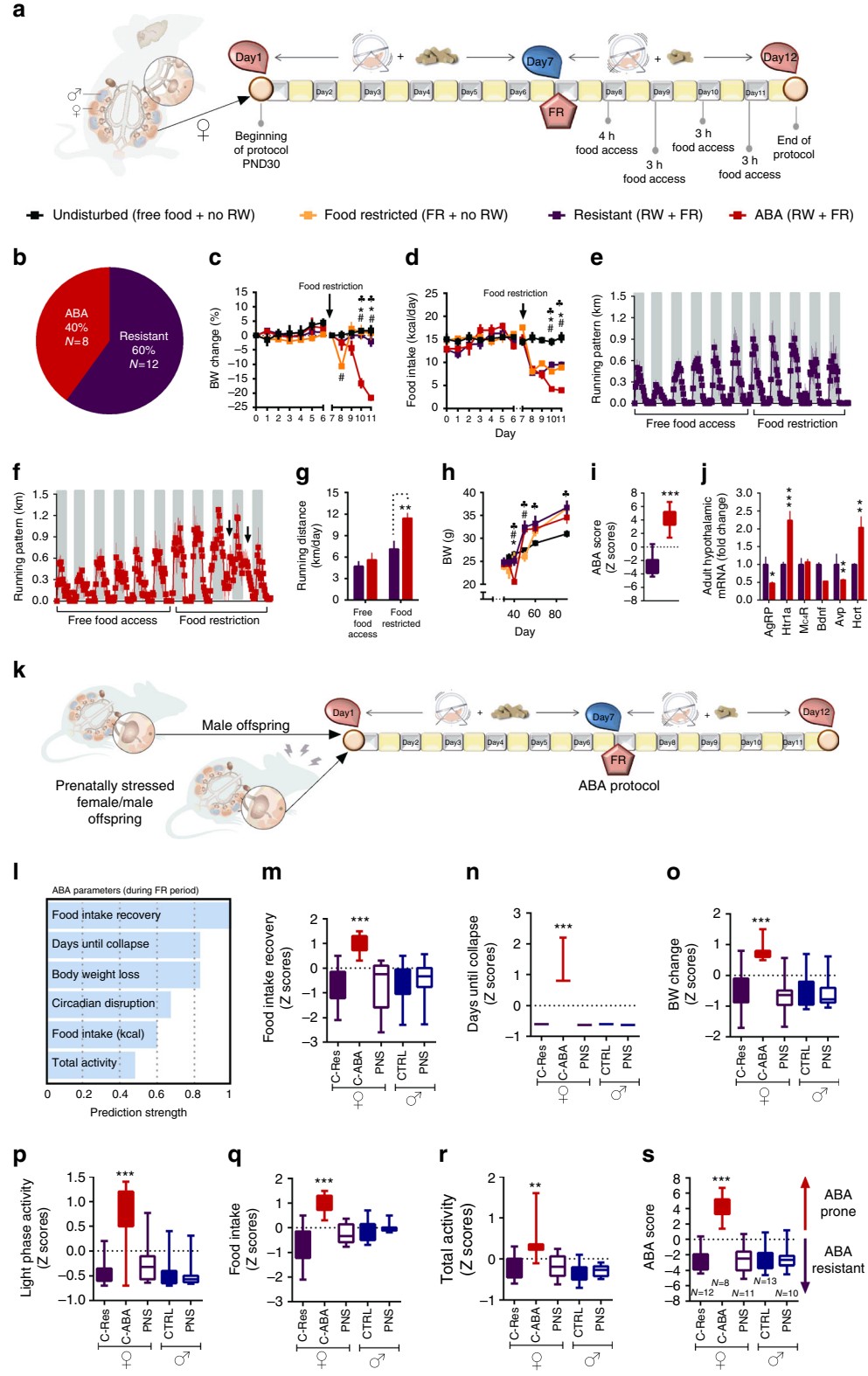

expression of the AN candidate genes in the hypothalamus of the fetuses according to placental miR-340 levels. We found that *AgRP* and *Hcrt* were highly expressed in control fetuses with high levels of miR-340 compared to fetuses with low levels of miR-340. In contrast, *Htr1a* and *Bdnf* were decreased (Fig. 3r).

**MiR-340 inversely correlates with placental amino acid transporters**. To gain insight into the potential effects of miR-340 levels on placental function and the uterinal environment that the fetuses were exposed to, we performed RNAseq in placentas with high and low endogenous miR-340 levels (Fig. 3m). Sleuth gene expression analysis revealed 881 upregulated and 779 downregulated genes in the high miR-340 group. Of these, we chose to focus on the solute carrier (SLC) group of membrane transport proteins, as well as on cholesterol transporters and the insulin-like growth factor (IGF) family given their crucial involvement in placental structure and function[35]. The two groups' profiles appear remarkably different in their capacity to allocate nutrients, as reflected by the different global expression in selected nutrient transporters (Fig. 4a). In particular, the levels of miR-340 inversely correlate with the expression of the amino acid transporters *Slc7a5c*, *Slc7a11*, and *Slc38a1*; the main lactate transporter *Slc16a3* and *Igf2* and *Igf2bp1* in mice (Fig. 4b). The Igf2 and the Igf family are heavily involved in nutrient transport and can further regulate thickness of the interhemel membrane connecting the maternal and fetal circulation[36]. Therefore, we examined the thickness of the junction in the two groups and found a thicker junctional area in the high miR-340 group, which was in accordance with reduced Igf2 levels[36] (Fig. 4c). When we explored the potential link between miR-340 levels and these transporters and growth factor, we found a potential pathway through the targets Nr3c1, Cry2, and H3f3b (Fig. 4d, based on STRING database https://string-db.org[37]). These findings suggest that the particular location of miR-340 in the junction may be linked to the regulation of nutrient transport to the fetuses, both by reducing fetal access to nutrients and amino acids and potentially through structural changes via Igf2. To examine the translational potential of our findings, we collected central, medial, and lateral human placental samples from healthy, full-term male and female pregnancies (see Supplementary Table 3 for the mothers' demographics and clinical characteristics). Analysis of the expression of miR-340 showed similar levels among boys and in the medial and lateral areas, but girls presented dramatically higher levels in the central

area, which is closest to the umbilical cord (Fig. 4e). Similarly to mice, human female placentas showed a higher variability in the expression of miR-340 and when correlated with the expression of the relevant transporters identified in mice, we found an inverse correlation between miR-340 and *SLC7A5*, *SLC7A11*, *SLC38A1*, *SLC16A3*, *IGF2*, and *IGF2BP1* (Fig. 4f, further transporters examined are presented in Supplementary Table 3). Thus, the results suggest that miR-340 may be involved in mediating the fetal acquisition of selective nutrients, thus affecting developmental programming of the offspring, both in humans and in mice.

**MiR-340 overexpression increases susceptibility to ABA**. With the aim of exploring whether high levels of placental miR-340 are directly related to increased vulnerability to ABA, we next generated placenta-specific transgenes[38,39]. Blastocysts were then infected with lentiviruses designed to overexpress (OE) miR-340 or with a control virus (CV) and were then transferred into pseudo-pregnant females for further development (Fig. 5a). As expected, infection of the blastocysts' trophoblast cells with lentiviruses led to placenta-specific infection (Fig. 5b) predominantly in the junctional area of the placenta (Supplementary Fig. 6). Validation of miR-340 expression in placentas infected with the miR-340 OE virus confirmed a threefold increase in miR-340 levels in both sexes. A similar increase was found in infected BeWo cells (Fig. 5c). Examination of fetal hypothalamic expression of AN candidate genes revealed a similar profile to wild-type females with high endogenous expression of placental miR-340, including upregulation of *AgRP* and downregulation of *Htr1a* (Fig. 5c and Fig. 3r).

We next explored the mRNA expression and protein levels of the targets both in the transgene placentas and in infected BeWo cells respectively. As expected, miR-340 OE generally inversely correlated with mRNA and protein of the targets in placental samples (Fig. 5d, f, h) and in BeWo cells (Fig. 5e, g, i). In a separate set of pregnancies, the transferred embryos were allowed to reach term. They remained untouched until 30 days old, when they underwent the ABA protocol (Fig. 6a). To determine susceptibility to ABA, we next calculated the Z scores of the ABA parameters according to the prediction strength of the original experiment (Fig. 1k). All raw data are presented in Supplementary Fig. 7. Accordingly, we found that the pattern of food intake recovery during FR was inefficient in miR-340 OE animals

---

**Fig. 1** Adolescent females show high susceptibility to activity-based anorexia (ABA), which is absent in males and is abolished by prenatal stress (PNS) in females. **a** ABA protocol experimental design. **b** Hierarchical cluster analysis split the females into ABA-prone and resistant to ABA. **c, d** The ABA protocol induced a dramatic decrease in body weight (BW) (repeated measures ANOVA $F(27,60) = 4.17$, $p = 0.000$ for time × treatment interaction) and food intake ($F(33,54) = 3.71$, $p = 0.000$ for time × treatment interaction) in ABA females compared to all other groups ($N = 6$ for undisturbed, $N = 6$ for FR, $N = 8$ for ABA and $N = 12$ for resistant). **e, f** Running pattern was normal in resistant females (**e**) and disrupted in the ABA group, which displayed high activity in the light phase (**f**). **g** Total running distance was significantly increased during food restriction (FR) in the ABA group ($F_{int(1,14)} = 7.33$, $p = 0.017$). **h** Long-term BW differed between the groups (repeated measures ANOVA $F_{int(15,100)} = 6.67$, $p < 0.0001$). All FR females weighed significantly more than undisturbed females. *$p < 0.05$ ABA vs. Resistant, #$p < 0.05$ ABA vs. FR and (black club) $p < 0.05$ ABA vs. undisturbed based on Tukey's multiple comparisons test. Data are presented as mean and s.e.m. **i** ABA animals scored significantly higher in the global ABA score compared to resistant females (Student's t test $t_{(18)} = 10.49$, $p < 0.0001$), which is composed of the sum of the Z scores of the six ABA parameters. Data presented in min. to max with median. **j** Hypothalamic gene expression of AN candidate genes differed between resistant and ABA females (MANOVA $F_{(6,5)} = 24.60$, $p = 0.001$). **k** To test the sex specificity and the contribution of PNS to ABA predisposition, we tested PNS offspring of both sexes together with control males on the ABA protocol using one-way ANOVA for each parameter with Tukey's multiple comparisons test. **l** ABA individual parameters for the males and PNS animals were calculated in Z scores according to each parameter's strength of prediction. **m−r** Control males and PNS males and females scored similarly low in food intake recovery ($F_{(4,49)} = 7.26$, $p < 0.0001$) (**m**), days until collapse ($F_{(4,49)} = 82.89$, $p < 0.001$) (**n**), BW change ($F_{(4,49)} = 9.81$, $p < 0.0001$) (**o**), circadian disruption (light phase activity)($F_{(4,49)} = 13.06$, $p < 0.0001$) (**p**), food intake ($F_{(4,49)} = 7.06$, $p < 0.0001$) (**q**) and total activity ($F_{(4,49)} = 8.34$, $p < 0.001$) (**r**) compared to resistant control (C-Res) females. **s** While control females split into resistant and ABA-prone subgroups, PNS females and all males were largely resistant to ABA ($F_{(4,49)} = 32.30$, $p < 0.0001$). RW running wheel, CTRL control. Data presented as min. to max with median

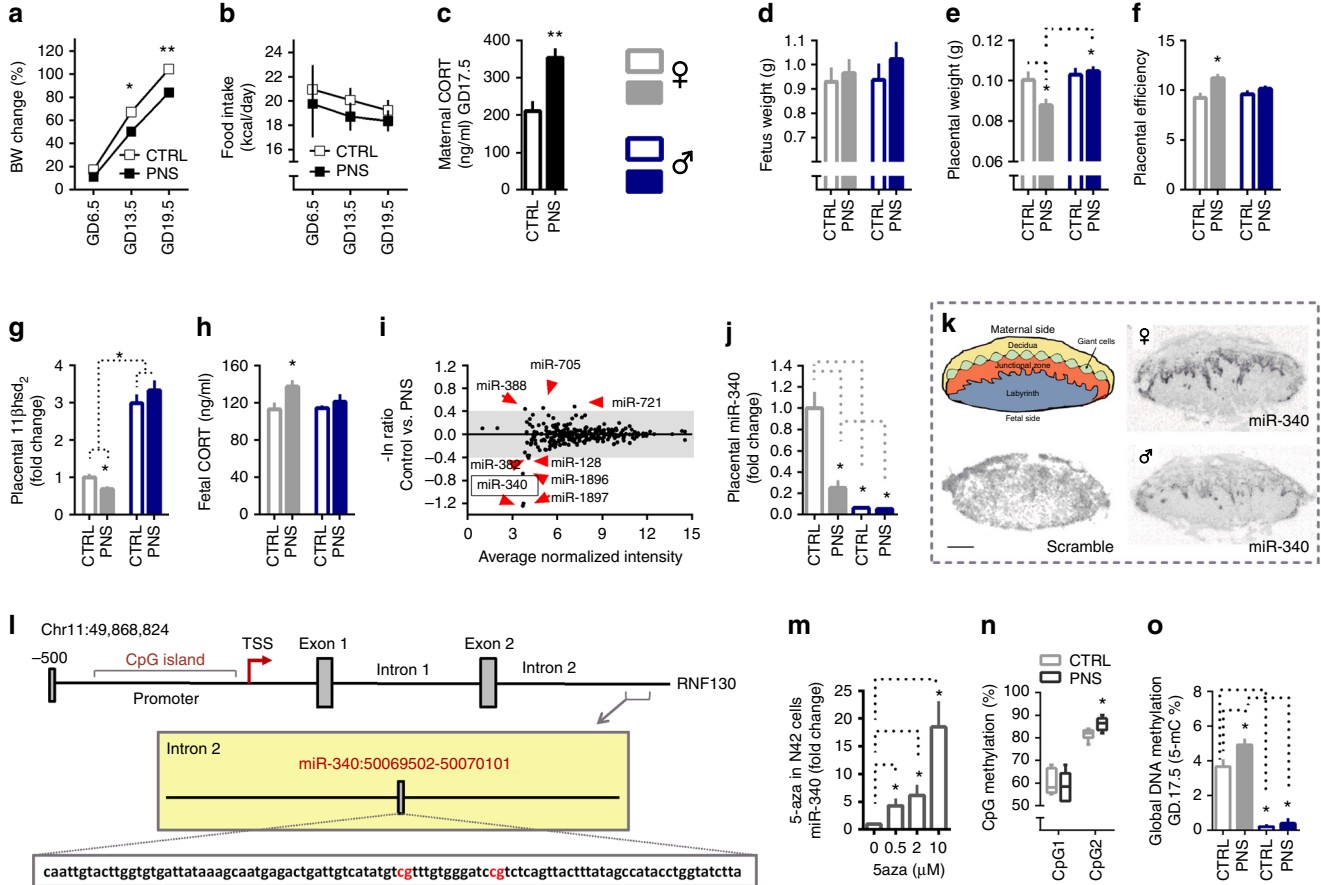

**Fig. 2** Prenatal stress (PNS) downregulates placental miR-340 through gene body DNA methylation. **a, b** PNS affected the pregnant dams' body weight (BW) (repeated measures ANOVA $F_{(1,9)} = 21.56$, $p < 0.0001$) without affecting food intake ($N = 5$–6). **c** PNS increased maternal corticosterone (CORT) ($t_{(10)} = 3.88$, $p = 0.003$). **d**−**f** Fetus weight was similar in control and PNS offspring (**d**), but placental weight was reduced by PNS only in females (two-way ANOVA $F_{int(1,18)} = 4.54$, $p = 0.047$) (**e**), resulting in increased placental efficiency (Kruskal−Wallis $H = 11.82$, $p = 0.008$, specific groups comparisons based on Mann−Whitney) (**f**). **g**−**h** Placental $11\beta hsd_2$ was overall higher in males (two-way ANOVA $F_{(1,46)} = 188.59$, $p = 0.000$) but was reduced by PNS only in females (interaction effect $F_{(1,46)} = 5.85$, $p = 0.020$, $N = 11$–12, from 5,6 different dams) (**g**) resulting in high fetal CORT in PNS females ($F_{sex(1,24)} = 5.28$, $p = 0.032$, $N = 12$) (**h**). **i** The placental miRNA array performed on females revealed downregulation of miR-340 by PNS. **j** MiR-340 validation showed reduced levels in PNS females and males of both prenatal conditions (two-way ANOVA $F_{sex(1,41)} = 41.00$, $p < 0.0001$) and ($F_{prenatal\ (1,41)} = 6.51$, $p = 0.015$). **k** MiR-340 is expressed in the junctional zone of the placenta. **l** Schematic representation of the *Rnf130* gene and CpG areas analyzed. **m** 5-aza treatment of N42 cells at increasing concentrations upregulated miR-340 levels (one-way ANOVA $F_{(3,15)} = 18.79$, $p < 0.0001$, with Bonferroni post-hoc tests). **n** CpGs on the miR-340 sequence were highly methylated and $CpG_2$ was significantly more methylated in PNS female placentas (Mann–Whitney $U = 3.50$, $Z = -2.35$, $p = 0.015$). Data are presented in min. to max. **o** Global DNA methylation differed dramatically among the groups (Kruskal−Wallis $H = 18.31$, $p = 0.000$). Females showed higher global methylation than males, an effect increased even further by PNS ($N = 6$, from 5 to 6 different dams). Significance between groups and sexes based on Tukey's multiple comparison tests and Mann−Whitney post-hocs when relevant. GD gestation day, CTRL Control. Data are presented in mean and s.e.m. Scale bar: 1 mm

(Fig. 6b). MiR-340 OE transgenes collapsed more frequently toward the end of the food restriction period (Fig. 6c), displayed higher BW change (Fig. 6d) and showed circadian disruption as reflected by the distance run (km) in the light phase (Fig. 6e) compared to transgene controls. While food intake during FR was not significantly affected (Fig. 6f), miR-340 OE induced hyperactivity in both males and females (Fig. 6g). Taken together, these parameters exposed the significantly higher susceptibility to ABA in miR-340 OE transgenes of both sexes, as reflected by the higher ABA score compared both to wild type (WT) and CV controls (Fig. 6h). Finally, hypothalamic assessment of AN candidate genes in these groups revealed long-lasting down-regulation of *AgRP* and upregulation of *Htr1a*, resembling the expression patterns found in the original control ABA group (Fig. 6i and Fig. 1j) and further confirming the central role played by the hypothalamic melanocortin and serotonin systems in anorexia. Altogether, these findings translated into higher

percentages of animals being prone to ABA as a result of miR-340 OE (Fig. 6j).

Taken together, we propose that placental miR-340, through the regulation of genes known to be involved in placental structure and function, such as *GR*, *Cry2* and *H3f3b* and downstream nutrient transporters, can change the fetal environment and hypothalamus to induce later life susceptibility or resistance to ABA (Fig. 7).

## Discussion

In the current set of studies, we have identified, for the first time, a potential mechanism of ABA gestational programming. It appears that early life variables that affect placental gene expression may play a crucial role in this predisposition. These findings are supported by the genetic epidemiology of AN, which indicates strong evidence for familial contribution but does not

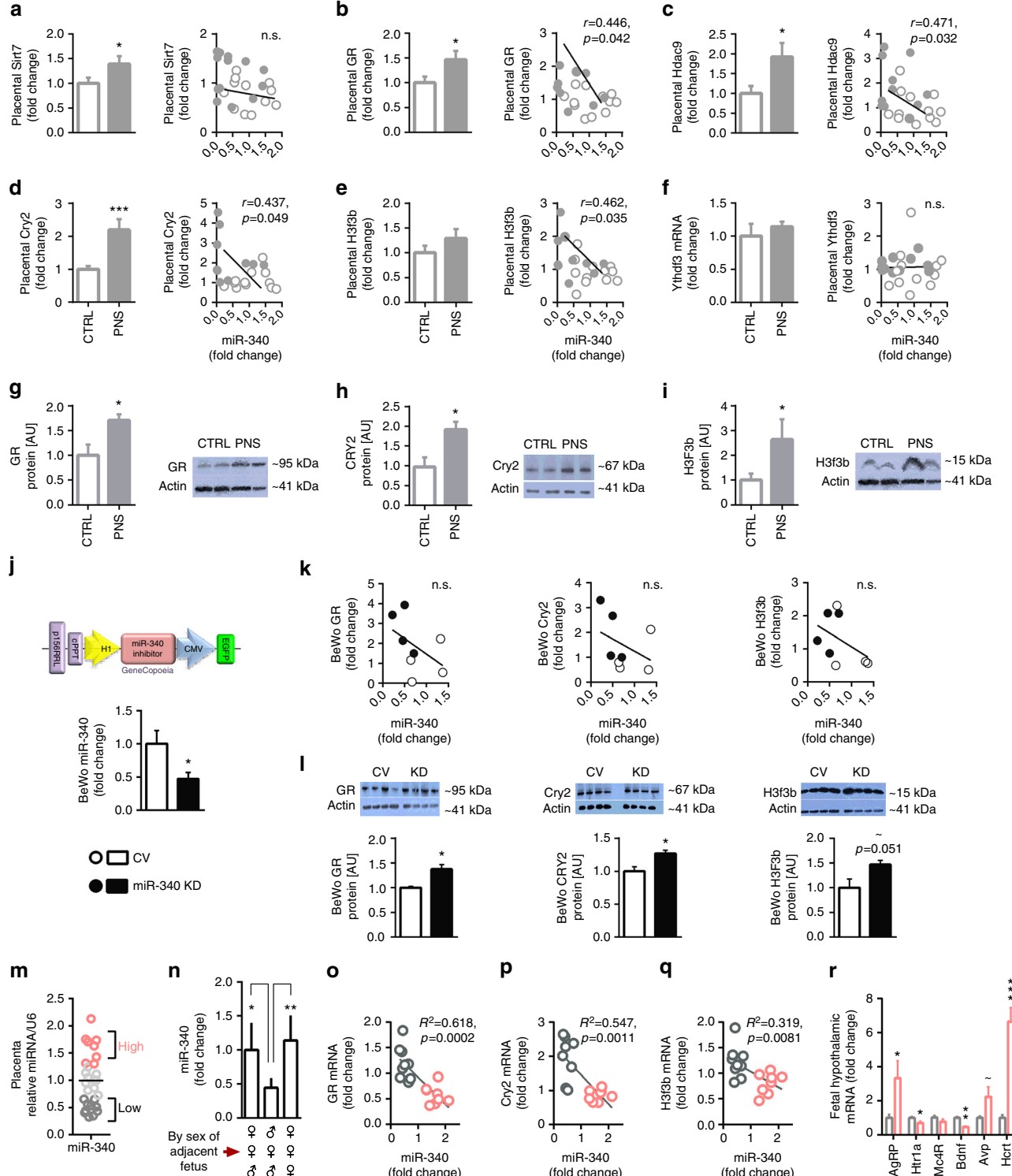

**Fig. 3** MiR-340 exerts its effects through targeting of GR, Cry2, and H3f3b. **a**–**f** Validation of potential miR-340 target in prenatal stress (PNS) vs. control female placentas showed that *Sirt7* and *Ythdf3* were not significantly correlated with miR-340 (**a**, **f**). **b**–**e** In contrast, *GR*, *Hdac9*, *Cry2*, and *H3f3b* expression was significantly higher in PNS and inversely correlated with miR-340. **g**–**i** Placental GR, CRY2, and H3F3b protein levels were increased by PNS ($N = 4$). **j** Lentiviral KD of miR-340 in BeWo choriocarcinoma cells reduced the expression of miR-340 ($F_{(1,7)} = 5.654$, $p = 0.050$, $N = 4$). **k** MiR-340 KD did not significantly affect mRNA levels of the target genes. **l** At the protein level, miR-340 KD increased GR ($t_{(6)} = 3.687$, $p = 0.010$) and CRY2 ($t_{(6)} = 3.156$, $p = 0.0197$), and tended to increase H3F3b. **m** Within the control group, miR-340 expression shows great endogenous variability ($N = 28$, from six dams). **n** Female fetuses surrounded by two males in the uterus show lower levels of placental miR-340 compared to females surrounded by females or mixed sexes (one-way ANOVA $F_{(2,23)} = 9.31$, $p = 0.0013$). Data are presented in mean and s.e.m. and Tukey's multiple comparison tests. **o**–**q** The "high" and "low" subgroups included samples above 1.3 and below 0.70 relative to the groups' average respectively. The expression of *GR*, *Cry2*, and *H3f3b* inversely correlated with the endogenous levels of miR-340. Data are presented as mean and s.e.m. **r** Fetal hypothalamic gene expression of AN candidate genes differed between the "high" and "low" subgroups ($F_{(6,5)} = 13.1$, $p = 0.006$). Data are presented in mean and s.e.m. ($N = 6$)

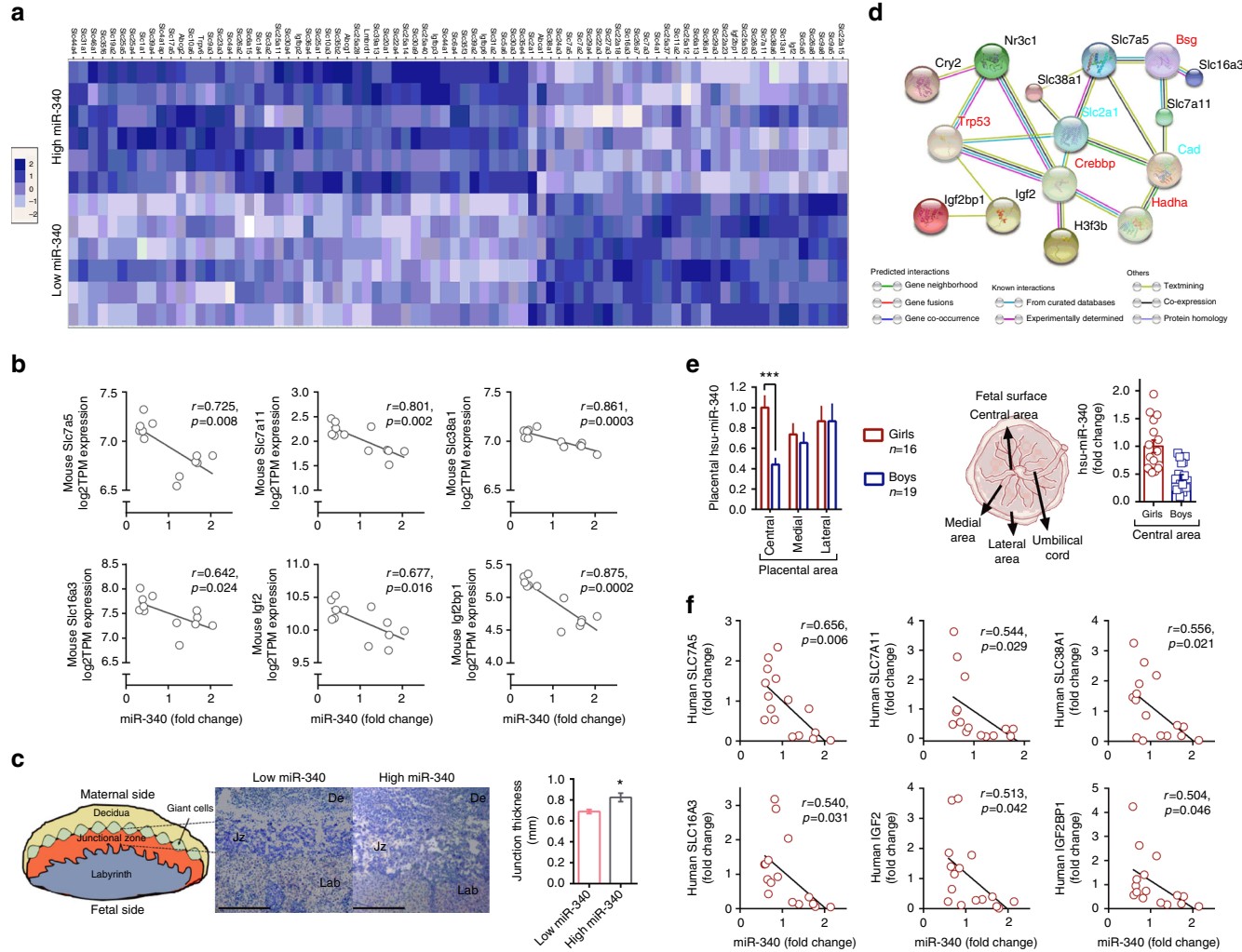

**Fig. 4** MiR-340 is associated with differential expression of nutrient transporters both in mouse and human female placenta. **a** Heat-map comparing placental nutrient transporters between female mice endogenously expressing high or low levels of miR-340. **b** The expression of *Slc7a5, Slc7a11, Slc38a1, Slc16a3, Igf2,* and *Igf2bp1* is inversely correlated with miR-340 in mice. **c** Eosin-hematoxylin staining of the mouse placenta show a thicker junctional zone in the high miR-340 group ($t_{(6)} = 3.07$, $p = 0.022$). **d** STRING pathway analysis linking the miR-340 targets *Nr3c1, Cry2,* and *H3f3b* with selected nutrient transporters (https://string-db.org). **e** Expression of miR-340 in the human placenta in central, medial, and lateral areas. Girls present higher levels in the central area compared to boys (two-way ANOVA $F_{(1,104)} = 14.78$, $p < 0.0001$). Data are presented as mean and s.e.m. **f** The expression of *SLC7A5, SLC7A11, SLC38A1, SLC16A3, IGF2,* and *IGF2BP1* are inversely correlated with miR-340 expression in the central area of human samples. Scale bar: 1 mm

always provide convincing evidence of hereditability according to population-based homozygous twin studies[40,41]. When it comes to gestational programming, the placenta plays a central part in shaping the fetal environment[42]. Placentas exhibit pronounced sexual dimorphism in response to variation in the maternal milieu both in humans[43] and mice[31,44], which appears to be the case for miR-340 in the junctional zone of the mouse placenta. This zone contains zygote-derived cytotrophoblasts, which have direct access to the maternal blood flow[45]. Two types of cytotrophoblasts exist in this zone at GD17.5 in mice, spongiotrophoblasts and trophoblast glycogen cells, both of which synthesize protein hormones such as IGF₂, a key growth and nutrient supply gene[46]. The human and mouse placentas share a similar function, but are structurally quite different. The corresponding region to the murine junctional zone in humans is the basal plate, which differently from mice presents multi-layered cytotrophoblastic columns with a morphological gradient of cells. Despite this difference, in both species this region does not contain any fetal blood, but is traversed by maternal blood

channels lined by zygote-derived trophoblast cells, through which maternal blood flows into and out of the fetal placenta/labyrinth[45].

The placental responses to stress include adaptations at multiple epigenetic levels, which affect molecular processes on a global scale. Abnormal DNA methylation is one of these processes[47], that in turn shapes placental morphology with far-reaching implications including impairments in placental development and nutrient supply. Maternal stress during pregnancy broadly influences the placenta, inducing the very well documented and varied negative effects in the offspring such as anxiety, depression, cognition, memory[48,49], and metabolism[50]. When PNS is limited to the last week of gestation, unlike our protocol, susceptibility to ABA appears to increase in a subset of rats[30]. A similar finding was recently reported for binge eating disorder using a different stress model but the same window of intervention[51]. Thus, ABA resistance resulting from chronic PNS may represent a beneficial side effect of an otherwise negative overall outcome.

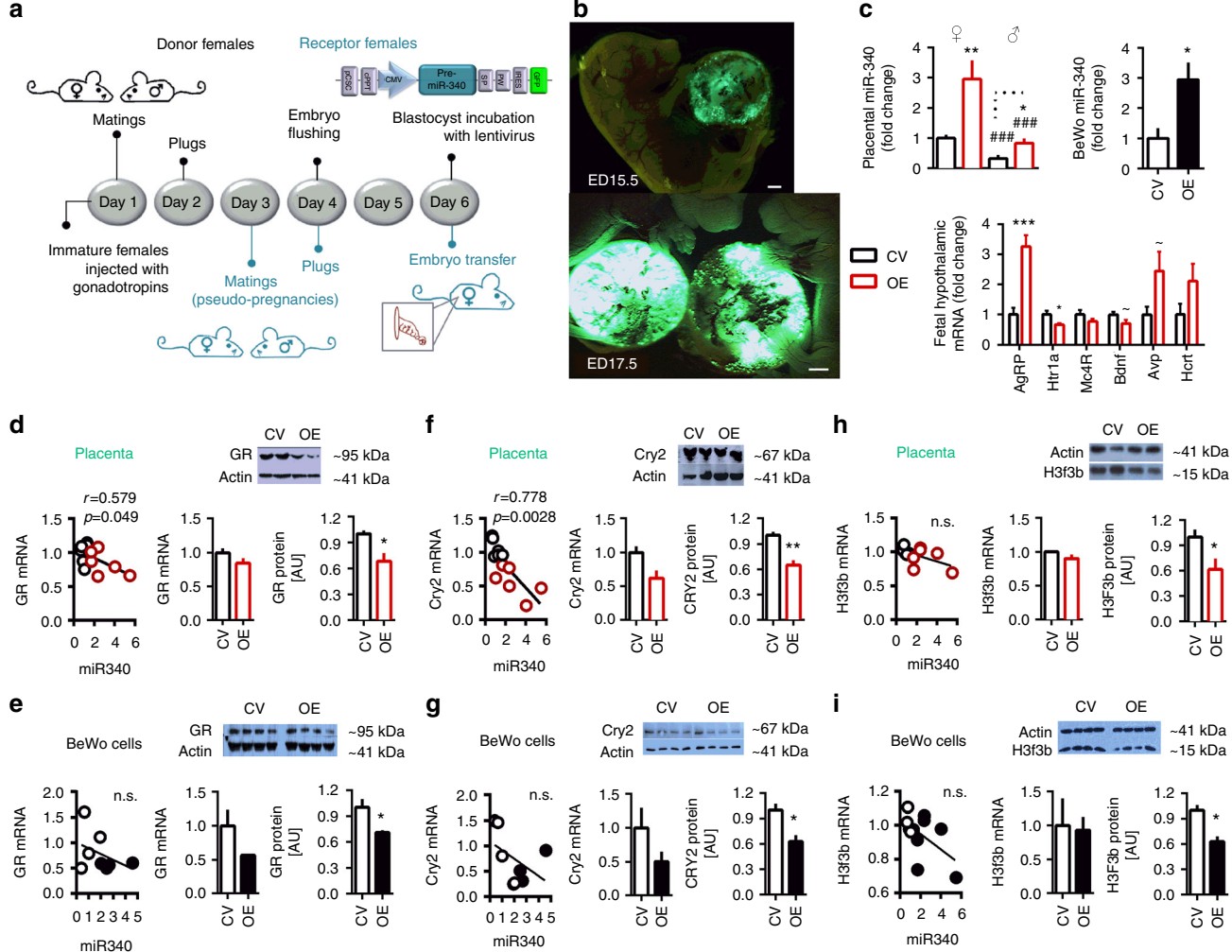

**Fig. 5** MiR-340 overexpression (OE) decreases GR, CRY2, and H3F3b protein in the placenta and in BeWo cells. **a** Experimental design for generation of placenta-specific transgenes. **b** Representative pictures out of about 20 transfers of placenta-specific infection. **c** Placental miR-340 was highly expressed in the female transgenes of both groups compared to males (two-way ANOVA $F_{sex(1,23)} = 47.39$, $p < 0.0001$) and was increased by the miR-340 OE lentivirus in both sexes ($F_{treatment(1,23)} = 28.78$, $p < 0.0001$, $N = 6$). Infection of BeWo cells with the miR-340 OE lentivirus led to increase in miR-340 expression levels (Unpaired $t$test $t_{(6)} = 1.77$, $p = 0.019$). Fetal hypothalamic gene expression of anorexia candidate genes differed between the CV and miR-340 OE groups (MANOVA $F_{(6,5)} = 7.08$, $p = 0.024$), with high expression of *AgRP* and low expression of *Htr1a*. **d** In miR-340 OE transgenes, placental *GR* mRNA inversely correlated with miR-340 ($N = 6$) and placental protein tended to be reduced ($t_{(6)} = 2.414$, $p = 0.052$, $N = 4$). **e** In BeWo cells, *Nr3c1/ GR* mRNA x miR-340 correlation showed a similar pattern to the placenta and miR-340 OE decreased GR protein levels ($t_{(6)} = 3.09$, $p = 0.011$, $N = 4$). **f** Placental *Cry2* mRNA inversely correlated with miR-340 ($N = 12$) and showed decreased protein levels after miR-340 OE (Unpaired $t$test $t_{(6)} = 2.451$, $p = 0.049$, $N = 4$). **g** In BeWo cells, miR-340 OE significantly decreased CRY2 protein levels ($t_{(6)} = 3.381$, $p = 0.0148$). **h–i** Mir-340 OE did not affect *H3f3b* mRNA expression but decreased H3F3b protein levels in the placenta ($t_{(6)} = 2.452$, $p = 0.0451$, $N = 4$) (**h**) and in BeWo cells ($t_{(6)} = 4.25$, $p = 0.0054$, $N = 4$) (**i**). Data presented as mean and s.e.m. CV control virus. Scale bar: 1 mm

Overexpression of GR, as seen in our placental PNS samples, induced multiple neuroendocrine changes that have been shown to result in a blunted response to restraint stress and to endotoxic shock in a different animal model[52]. Accordingly, high maternal cortisol in utero and/or inhibition of $11\beta HSD_2$ are associated with programmed outcomes in childhood including higher blood pressure, behavioral disorders, and altered brain structure[53]. Given the remarkable resistance of males to ABA, a potential mechanism of ABA prevention in females by PNS may be related to its well-known masculinizing effects. PNS increases anogenital distance, which is a sexually dimorphic biomarker of prenatal androgen exposure in many species[54]. We noticed that females were undistinguishable from males in the PNS group at birth due to larger than usual anogenital distance. This effect resembles prenatal testosterone exposure, which was shown to masculinize

the food intake pattern of female offspring[55]. Moreover, disorganized eating in girls may be improved by prenatal androgen exposure, as shown in a twin study in which female pairs had significantly higher disordered-eating scores than girls with male twins[56]. This "prenatal androgen exposure hypothesis" has been more frequently linked to autism spectrum disorders[57], but could also be involved in determining the level of predisposition to ABA in females. This is particularly likely given the well-documented effects of uterine position on the fetuses: females developing between two males are more likely to show masculinized behavioral, anatomical physiological traits due to the transfer of testosterone from the adjacent fetuses[58]. This could be a potential source of the great variability adolescent WT females show in their response to ABA. Thus, females surrounded by other female fetuses may be more vulnerable to ABA as suggested by human

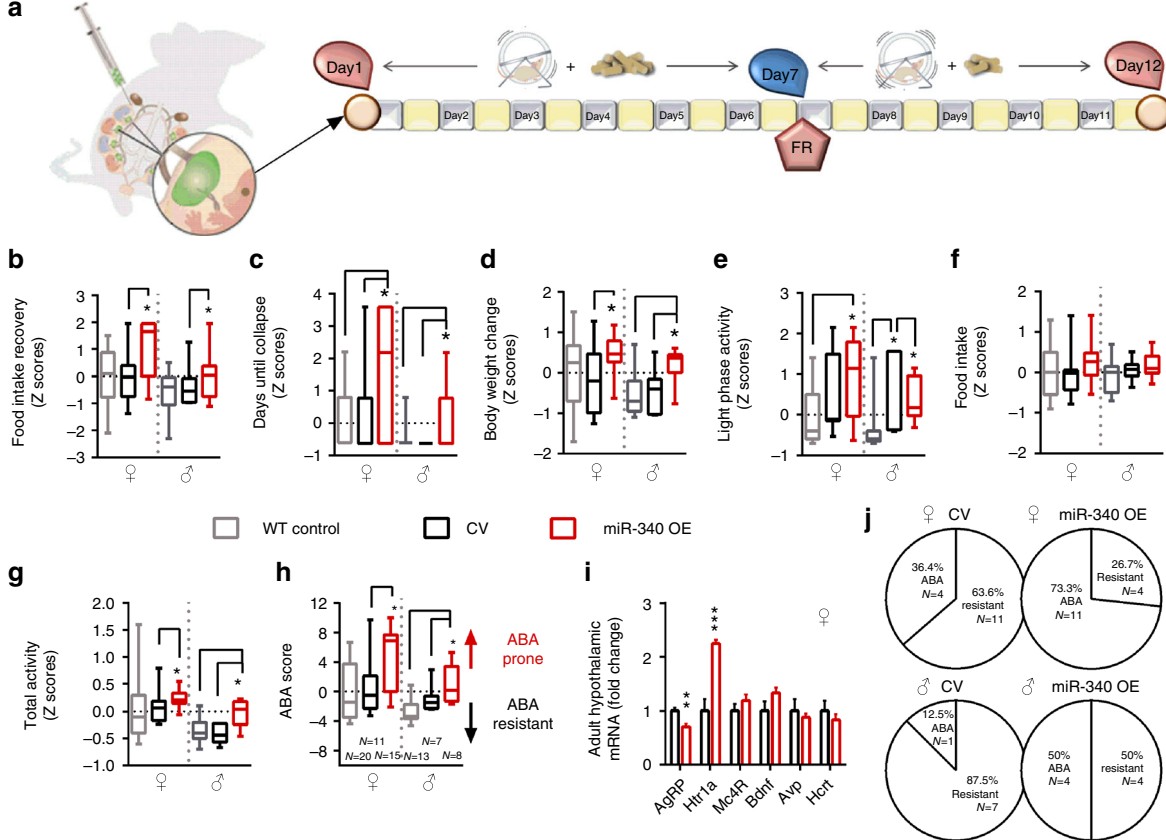

**Fig. 6** Placental miR-340 overexpression (OE) robustly increases the susceptibility to activity-based anorexia (ABA) in both sexes. **a** Transgene animals of both sexes were exposed to the ABA protocol. **b**−**g** MiR-340 OE transgenes of both sexes scored higher than controls in food intake recovery (one-way ANOVA $F_{(5,68)} = 4.49$, $p = 0.0014$) (**b**), days until collapse ($F_{(5,68)} = 5.36$, $p = 0.0003$) (**c**), body weight (BW) change ($F_{(5,68)} = 3.99$, $p = 0.003$) (**d**), circadian disruption (light phase activity) ($F_{(5,68)} = 5.01$, $p = 0.0006$) (**e**) and total activity ($F_{(5,68)} = 6.81$, $p < 0.0001$) (**g**) during the food restriction part of the protocol. In contrast, food intake during food restriction did not significantly differ between the groups (**f**). **h** Altogether, placental miR-340 OE robustly increased the susceptibility to ABA in both sexes compared to both WT controls and CV animals ($F_{(5,68)} = 6.83$, $p < 0.0001$). Data presented as min to max with median. Differences between the groups are based on Tukey's multiple comparisons test. *$p < 0.05$. **i** Hypothalamic gene expression of anorexia candidate genes differed between CV and miR-340 OE females (MANOVA $F_{(6,5)} = 53.83$, $p < 0.000$), with low expression of AgRP and high expression of Htr1a. Data are presented in mean and s.e.m. ($N = 6$). **j** MiR-340 OE increased the susceptibility to ABA in both sexes. CV control virus

twin studies[56,59]. While the sex-specific physiological mechanism through which placental miR-340 methylation and expression is determined under both stressful and normal conditions remains unclear, hormonal alterations in utero may represent a potential candidate as suggested by Rnf130 regulation by LH/hCG in Leydig cells in rats[60]. Finally, the fact that ABA and AN are typically triggered during adolescence may further suggest that the prenatal hormonal effects on female offspring may form a platform for a later triggering or protective effect of gonadal steroids on this phenotype.

The modulation of the targets identified here, and potentially dozens of further targets by miR-340, likely affect nutrient transporters and growth factors, which may have in turn a pivotal influence on fetal development and programming. The ability of a single miR molecule to affect the functional levels of multiple proteins/pathways supports therefore their possible involvement in complex disorders, such as EDs, which are suggested to be polygenic by nature and influenced by gene × environment interactions[61]. Similarly to miRs, transcription factors like GR[62] are highly involved in the regulation of gene expression. GR is widely expressed both in the labyrinth and the junctional zone of the murine placenta[63]. It is involved in the expression and alteration of a great number of placental genes including the regulation of trans-placental glucose transfer, through

modulation of the glucose[64] and amino transporters. It is therefore a crucial factor for sustaining fetal life and securing its normal development[65] affecting the newborns' neurobehavior[66]. GR interacts with Cry2, a key component of the circadian core oscillator complex that regulates the circadian clock and has been detected both in the labyrinth and the junctional zone in rats[67]. The clock genes are widely involved in the regulation of placental function, suggesting that normal circadian variations may be involved in a healthy placental phenotype[67]. Finally, H3f3b, one of the replication-independent variants of the Histone 3 mammalian isoform, is enriched in coding regions and at specific chromatin landmarks in mouse somatic and embryonic cells[68]. While its role in placental structure and functioning is largely unknown, it is suggested to affect chromatin architecture changing the regulation of epigenetic mechanisms during development[69]. Given their function, it is not surprising that the miR-340 targets may affect the transfer of nutrients to the fetus by regulating selective transporters. In particular, less transfer of amino acids to the fetus, as reflected by lower expression of three key transporters, and Igf2, which can affect the thickness of the junction hindering the transfer of nutrients[70].

Overall, we have identified a placental mechanism through which miR-340 mediates vulnerability to ABA, likely through influencing placental function. MiR-340 presents clear sexual

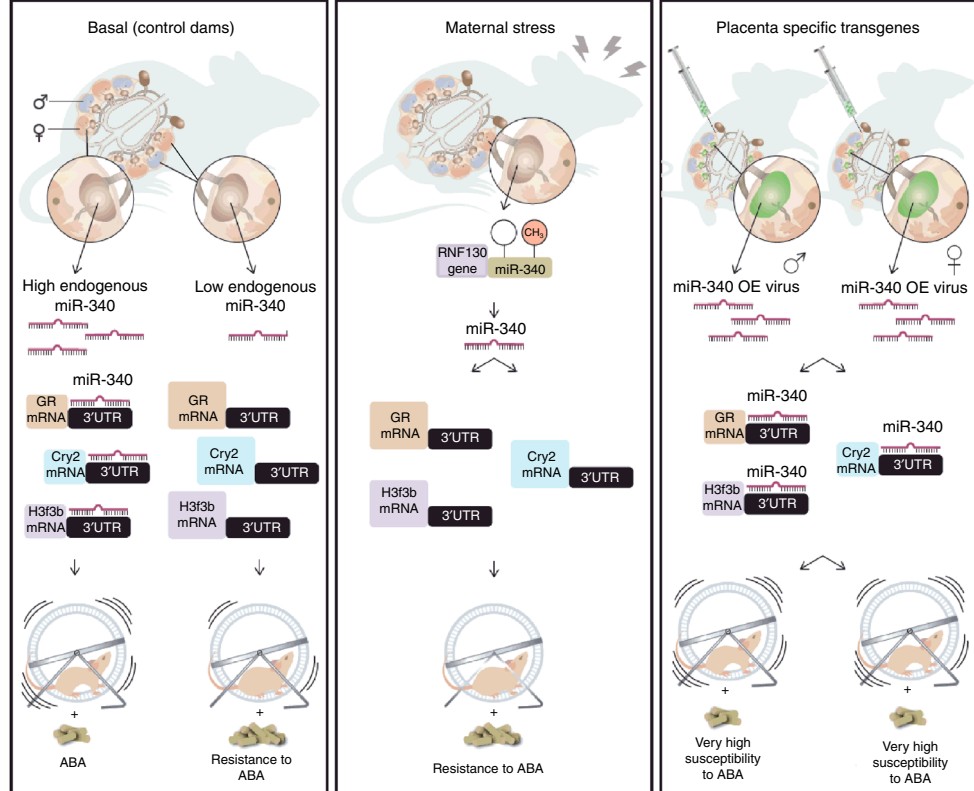

**Fig. 7** Summary scheme comparing placental miR-340 expression and the adult outcome in the different experiments. Female offspring born from control dams present high levels of placental miR-340, while the male levels are endogenously low. MiR-340 targets *Nr3c1/GR*, *Cry2*, and *H3f3b* programming susceptibility to activity-based anorexia (ABA) specifically in females (left panel). Prenatal stress downregulated miR-340 through gene-body DNA methylation, upregulating *Nr3c1/GR*, *Cry2*, and *H3f3b*, thus are protecting the female offspring from sensitivity to ABA (central panel). Upregulation of placental miR-340 through lentiviral infection of blastocysts reduced GR/CRY2/H3F3b protein levels, and robustly increased ABA susceptibility in both sexes (right panel). CV control virus, GR glucocorticoid receptor, OE overexpression

dimorphism and is downregulated by PNS inducing a "protective" effect against ABA. Placenta-specific OE of miR-340 in transgene mice exposed its role in the regulation of placental functioning through GR/Cry2//H3f3b and its involvement in gestational programming of ABA, potentially influencing fetal brain development. Our findings directly link, for the first time, the variability in the expression of a placental miR with fetal programming and later vulnerability to AN and provide important insights into the ontogeny of this poorly understood ED.

## Methods

**Animal care**. ICR/CD1 mice (Harlan Sprague Dawley Inc., Indianapolis, IN) were maintained in a pathogen-free temperature-controlled (22 ± 1 °C) mouse facility on a reverse 12 h light−dark cycle at the Weizmann Institute of Science, according to institutional guidelines. Food (Teklad Global, Harlan Sprague Dawley Inc., Indianapolis, IN) and water were given ad libitum (apart from during FR). All experimental protocols were approved by the Institutional Animal Care and Use Committee of the Weizmann Institute of Science.

**Breeding**. Female ICR mice were mated at 11−13 weeks of age. Two or three females were housed with one male (minimum age 12 weeks) at the beginning of the dark period and were examined for the presence of a vaginal copulation plug at the end of the dark period. Presence of a copulation plug denoted day 0.5 of gestation. After breeding, the females were individually housed.

**Experimental groups**. Females with plugs were randomly assigned to the control or PNS treatment. From day 18.5 of gestation, females were checked twice a day for the presence of a litter (0900−1000 hours, 1700−1800 hours). Newborn litters found by 1800 hours were designated as born on that day—postpartum day 0 (PPD 0). On PPD 1, pups were counted and litters were culled to ten pups (with sex

distribution kept as equal as possible in each litter). Litters with less than seven pups were excluded.

**PNS protocol**. From gestation day (GD) 1.5 until GD16.5, pregnant females were exposed to a chronic variable mild stress protocol, including two short manipulations during the dark phase and a further overnight (ON) manipulation during the light phase. The chosen stressors did not induce pain and did not directly influence maternal food intake or weight gain and were repeated on a weekly basis. The short stressors included multiple cage changes, cage tilt, white noise, water in cage or no bedding for two periods of 2 h, immobilization in a tube or elevated platform for 30 min and swim stress in warm water for 15 min. Overnight manipulations included illumination, saturated bedding (with water), novel object (marbles), and overcrowding.

**Activity-based anorexia model**. On PPD 21−22, pups were weaned and offspring were group housed until PPD 30, where they were randomly designated to one of two experimental groups (using a randomized complete block design to avoid potential litter effects): (1) Control group (free food access, no RW); (2) Food-restricted group (FR group): Free food the first week, FR the second week, no RW; (3) RW group (free food access in the first week, FR in the second week, free RW). On PND 30, which is considered the onset of adolescence in mice as determined by the time of the first estrous[70], ICR mice in the ABA groups were placed in an RW system in order to record running pattern (diameter 115 mm, TSE-Systems, Bad Homburg, Germany). The protocol lasted for 11 days during which food intake and BW were measured daily. The first week was defined as a "training period", during which food and water were given ad libitum. On the first day of the second week food access was limited to 4 h, and on the following days the food access was limited to 3 h. Animals were defined as anorexic in retrospect by cluster analysis according to the following criteria, which was based on previous studies[21–24] and our preliminary findings: (1) Percentage weight loss; (2) Food intake during FR; (3) Food intake recovery (%) during the FR period; (4) Circadian disruption (km ran during the light phase); (5) Days until collapse (faint and become hypothermic) and (6) Total running distance (km) when FR. Animals that developed ABA were

removed from the RW cage once they collapsed and were subsequently given food ad libitum and monitored until fully recovered. The predictive strength of the six ABA parameters was determined by two-step cluster analysis and was used to calculate the Z scores.

**Production of lentiviral vectors.** MiR-340 was PCR amplified from mouse genomic DNA, using the following oligonucleotide primers, both carrying restriction sites for AgeI :5′-TCTTCAAGAGGGAGCCTCAG-3′ and 3′-AACAGGCATGATCTGTGGTG-5′. The purified PCR product (~175 bp) was digested using AgeI enzyme and ligated to pCSC-SP-PW-IRES-GFP plasmid. MiR-340 sequence was confirmed by DNA sequencing. High titer lentiviruses expressing miR-340 and green fluorescent protein (GFP) were produced by transient transfection in HEK293T cells. Infectious particles were harvested at 48 and 72 h post-transfection, filtered through 0.45 µm-pore cellulose acetate filters (Sigma-Aldrich), concentrated by ultracentrifugation, re-dissolved in sterile HBSS, aliquoted and stored at −80 °C. Vector concentrations were analyzed using eGFP fluorescence in HEK293T cells infected with serial dilutions of the recombinant lentivirus[71,72].

**Generation of placenta-specific lentiviral system.** Three-week-old female mice were hyper-ovulated with gonadotropins (PMSG) (Sigma-Aldrich) followed 48 h later by hCG (5 IU each) and were then mated with males. Two days after the detection of a copulation plug, oviducts were dissected (Embryonic day (ED) 2.5) and flushed with M2 medium (M7167, Sigma-Aldrich, St. Louis, MO), using a syringe with 30-gauge blunt hypodermic needle. Embryos at their morula stage (ED2.5) were collected and rinsed through several drops of M2 medium. The zona pellucida was then removed by rinsing with Tyrode's solution (M8410, Sigma-Aldrich, St. Louis, MO). Denuded embryos were re-washed with M2 medium and placed in 50 µl drops of M16 medium (M7292, Sigma-Aldrich, St. Louis, MO) on a 35-mm tissue culture plate. They were then incubated for 2 nights at 37 °C and 5% $CO_2$ for further development. Embryos at their blastocyst stage (ED4.5) were then incubated in 30 µl drops of M16 + 3 µl concentrated lentivirus for 6 h. Using a stripper micropipette (Cat# MD-MXL3-SRT-CGR; Zotal Ltd, Tel Aviv, Israel), infected blastocysts were then transferred into fresh drops of M2 medium and then into ED2.5 pseudo-pregnant females' uterus for further development[38,39]. This technique is highly specific to the transduction of the junction, therefore making vesicular transfer highly unlikely.

**Generation of pseudopregnant females.** The pseudo-pregnant females were produced by mating with sterilized males. Vasectomies were performed on 2-month-old males by making an incision across the lower abdomen, at the midline of the scrotal sac, under Ketamine:Xylazine (1:1 20% in saline) anesthesia. Then, a section of the vas deferens (0.5 cm) was bilaterally removed. Males were allowed 2 weeks to recover and were mated with females to confirm sterilization before the beginning of the protocol.

**Embryo tissue collection.** On GD17.5, pregnant females were anesthetized with an overdose of Ketamine-Xylazine (1:1, 20% in saline) and through cesarean section both the embryos and placentas were excised. Embryos were weighed and decapitated, the brain was extracted and the hypothalamus was removed, immediately frozen on dry ice and stored at −80 °C. Trunk blood was collected in EDTA-coated tubes (MiniCollect, Greiner bio-one, Austria). Placentas were also weighed and immediately stored at −80 °C. A portion of the embryos' tail was removed for later DNA extraction and sex genotyping. For analyses comparing between groups, only one fetus per litter (per sex) was used. These were chosen from the central uterinal location, with one male and one female as adjacent fetuses. Males chosen had one male and one female as adjacent fetuses. When the focus was intra-litter variability, the fetuses located at the extremes of the uterus were excluded.

**Adult tissue collection.** Adult animals were killed at the age of 3 months, 4 weeks after the end of the ABA protocol and when fully recovered. After decapitation, the hypothalamus was dissected and immediately frozen. Trunk blood was collected, immediately centrifuged and plasma samples were and stored at −80 °C.

**Genotyping.** Sex of the embryo was determined by Sry genotyping (forward 5′-TCATGAGACTGCCAACCACAG-3′ and reverse 5′-CATGACCACCACCAC-CACCAA-3′)[73]. Amplification product was detected by 2% gel electrophoresis.

**Offspring physiological measurements.** Pregnant and lactating dams underwent weekly follow-up of body weight and food intake. For pre-weaning weights, total litter weights were averaged (males and females apart). Post-weaning, pups were weighed individually and food intake was monitored weekly.

**Plasma corticosterone measurement.** Corticosterone levels were analyzed using radioimmunoassay (MP Biomedicals, Solon, Ohio, USA).

**Mouse RNA extraction and real-time PCR.** For placental tissue, purification of total RNA containing miRNAs and DNA was done using Tri Reagent (MRC Inc., Cincinnati, OH) according to the manufacturer's recommendations. gDNA was removed using Turbo DNAse digestion in solution, followed by heat deactivation (#AM2238, Thermo Fisher Scientific). Hypothalamic RNA was extracted using the miRNeasy Mini Kit (#217004, Qiagen) in conjunction with on column digestion of gDNA using Turbo DNAse (see above). RNA preparations were reverse transcribed to generate cDNA using miScriptII Reverse transcription kit for miRNAs/mRNA (QIAGEN, #218160, Hilden, Germany). Quantitative mRNA/miRNA expression were done using a SYBR®Green PCR kit (QIAGEN, Hilden, Germany) and miScript SYBR (#204057 QIAGEN, Germany) respectively, according to the manufacturer's guidelines and a StepOnePlus® Real time PCR system (Applied Biosystems, Waltham, MA), using specific primers. U6 snRNA was used as internal control for the miRs and Tbp was used as internal control for mRNA in placental samples.

**MicroRNA Agilent array.** MiRNA array of the placenta was performed using Agilent's microarray (SurePrint G3 Unrestricted GE 8*60k, G4872A; Agilent Technologies, Santa Clara, CA). The bioinformatic analysis was performed using the Limma (Liniar Models for MicroArray Data) package which is part of the Bioconductor software (http://bioconductor.org). We corrected the background with the Normexp method, used the Loess method for the within arrays normalization and the Aquantile for between arrays normalization. A linear model was applied in order to find the differentially expressed genes (a simple Bayesian model). Gene ontology analysis was performed using WebGestalt 58. (http://bioinfo.vanderbilt.edu/webgestalt/) (Supplementary Data 1).

**Taqman array.** Placental gene expression of potential miR-340 targets was analyzed by a Taqman fast small RNAs assay (Applied Biosystems, Waltham, MA) (Supplementary Table 1).

**Mouse and human RT-PCR primers.** See Supplementary Tables 4 and 5.

**5-aza functionality assay of methylation.** N42 immortalized hypothalamic cells were plated into 35 mm plates and treated for four days with different concentrations (0, 0.5, 2, 10 µM) of 5-Aza-2′-Deoxycytidine (Sigma-Aldrich, St. Louis, MO). Every day in the evening, medium was replaced with fresh growth medium containing fresh 5-Aza. Cells were then lysed and RNA was extracted using the miRNeasy mini kit (QIAGEN, Hilden, Germany). RNA was reverse transcribed to cDNA using miScript Reverse transcription kit (QIAGEN, Hilden, Germany).

**Global DNA methylation.** Global methylation of the placenta was done using the MethylFlash Methylated DNA Quantification ELISA kit (Epigentek Group Inc., Farmingdale, NY) according to the manufacturer's instructions.

**DNA methylation of Rnf130 promoter and miR-340 sequences.** Methylation analysis by pyrosequencing of bisulfite-treated genomic DNA was performed by Varionostic GmbH (http://www.varionostic.de/). Genomic DNA was bisulfite converted using the EZ DNA Methylation Gold Kit (Zymo Research, Irvine, CA). Amplicons were generated from bisulfite DNA covering 134 CGs in the CpG island located in the promoter region of the Rnf130 gene and the sequence coding for miR-340 on intron 2 (Fig. 2 and Supplementary Fig. 3). Sequencing was performed on the Q24 System with PyroMark Q24 analysis software in CpG (QIAGEN, Hilden, Germany). Data were analyzed using SPSS Statistics, Version 17 (SPSS Inc., Chicago, IL) as indicated.

**Western blot.** Placental protein was purified in RIPA buffer, centrifuged for 10 min at 4 °C, separated in a 10% polyacrylamide gel by electrophoresis and transferred onto nitrocellulose membranes. The transfer was performed at 100 v, 350 mAmp for 1 h. After washes with PBST (PBS + 20% Tween20), membrane was blocked with 10% skimmed milk in PBST for 1 h. The primary antibodies anti-H3F3B (1:10,000, #GTX115549 GeneTex, Irvine, CA), anti-NR3C1 (GR) (1:200, (M-20) sc-1004; Santa Cruz Biotechnology), anti-CRY2 (abcam 1:500 ab38872), anti HDAC9 (1:500, B-1, sc-398003, Santa Cruz Biotechnology) and anti-Actin (1:1000, sc-1616; Santa Cruz Biotechnology) were added to PBST and placed on constant shaking at 4 °C overnight. After several washes with PBST, the secondary antibodies (Cell Signaling Technology, Beverly, MA) were added for 1 h incubation at RT. Finally, the membranes were visualized using ECL (Thermo Fisher Scientific, Waltham, MA) and film (Fujifilm, Tokyo, Japan). For tests and membranes, see Supplementary Fig. 8.

**Gross placental structure analysis.** The thickness of the placental section occupied by the junctional zone (Jz) was determined on an M205C stereo microscope (Leica, Wetzlar, Germany) and the Leica Application Suite X software (LAS X) following eosin−hematoxylin staining by averaging three measurements in the central area ($N = 4$).

**Libraries and RNAseq**. Libraries were prepared with the Illumina TruSeq Stranded Total RNA Library Preparation kit with Ribo Zero Gold (Illumina, #RS-122-2301) according to the instructions, using 1000 ng total mouse placenta RNA as starting material. Libraries were quantified on a Qubit fluorometer and by qPCR with a KAPA Library Quantification Kit for Illumina libraries (# KK4828). Size distribution was checked using the Agilent High Sensitivity DNA Assay (#5067-4626) on an Agilent Bioanalyzer. Samples were denatured using 1 N NaOH, diluted to a concentration of 3 nM with ExAMP mastermix and loaded onto a HiSeq 4000 machine (Illumina, San Diego, CA; #SY-401-4001) with 1% PhiX control (Illumina, #FC-110-3001) spiked in. HiSeq 3000/4000 flow cells and HiSeq 3000/4000 SBS sequencing chemistry were used for paired-end sequencing with a read length of 100 bp for each direction. Sequencing was performed at the Helmholtz Center (Munich, Germany).

**RNAseq analysis**. The quality of sequencing reads was verified using FastQC 0.11.5 (http://www.bioinformatics.babraham.ac.uk/projects/fastqc). Adapters were trimmed using cutadapt v.1.9.1[74] in paired-end mode. For quantification of gene expression, kallisto 0.43.1[75] was employed using the mouse Ensembl annotation v79 (downloaded from http://bio.math.berkeley.edu/kallisto/transcriptomes/). The 100 bootstaps sleuth 0.28.1[76] was used for the analysis of differentially expressed genes in gene aggregation mode, requiring $q$-values of <0.05 for both Wald and likelihood ratio tests, with a beta value cut-off 0.25. For KEGG pathway analysis, GAGE 2.24.0[77] was used with log2 transformed, filtered and normalized TPM values extracted from sleuth. An FDR of 0.1 was applied as cut-off for significance.

**BeWo cell culture**. BeWo choriocarcinoma immortalized cells were plated into 35 mm plates with F12 medium with 2 mM L-glutamine, fetal bovine serum (10%) and penicillin streptomycin solution (10,000 units/ml Penicillin GSodium Salt, 10 mg/ml Streptomycin Sulfate) (Thermo Fisher Scientific, Waltham, MA, USA) at 37 °C in a 5% $CO_2$ humidified incubator. BeWo cells were treated for 48 h with lentiviral vectors to overexpress or knockdown miR-340. Cells were then lysed and RNA was extracted using the miRNeasy mini kit (QIAGEN, Hilden, Germany) or were lysed using RIPA buffer for protein extraction.

**Body composition**. Body composition was assessed using Echo-MRI-100™ (Echo Medical Systems, Houston, TX, USA).

**Metabolic assessment**. Indirect calorimetry, food and water intake were measured using the LabMaster system (TSE-Systems, Bad Homburg, Germany). The LabMaster instrument consists of a combination of sensitive feeding and drinking sensors for automated online measurement. The calorimetry system is an open-circuit system that determines $O_2$ consumption, $CO_2$ production, and respiratory exchange ratio. Data were collected for five consecutive days after 24 h of adaptation for the apparatus.

**Maternal behavior**. On days 6/7 and 17/18 postpartum, patterns of undisturbed nocturnal maternal behavior were observed during 160 min sessions. Each mother was observed every 15 min, for 1−3 s. This allowed the identification of the ongoing maternal behavior at the observation time. Various maternal and non-maternal behaviors were recorded in every observation. The score was "1" if the behavior occurred and "0" if it did not occur. Maternal behavior measures were based on existing literature[78] and included both self-(grooming, eating) and pup-directed behaviors (nursing, licking/grooming) and activity measures.

**Human placental tissue collection**. Placental tissues were collected after obtaining informed consent from pregnant women at the Division of Gynecology and Obstetrics of the Lindenhofgruppe, Bern, Switzerland. The studies were approved by the cantonal ethic commission of the canton of Bern (KEK), # 2016-00250. Term placentas (38−40 weeks) were collected from uncomplicated pregnancies after elective cesarean section without prior labor symptoms upon patients' request or due to breech presentation ($n = 35$) (Supplementary Table 2). Placental tissues were collected from the villous tree within 1 h of delivery. To minimize blood contamination, each piece of tissue was intensively washed in Dulbecco's phosphate-buffered saline. Tissue samples were then immediately snap-frozen in liquid nitrogen and stored at −80 °C until RNA isolation. As placenta is a heterogeneous tissue, and physiological differences in gene expression may occur depending on the sampling site, we standardized the sampling protocol and routinely collected tissue samples from three different locations within the placenta: central (C), paracentral/medial (M) and peripheral/lateral (L) parts. Total RNA was isolated similarly to the mouse samples, using cold Trizol reagent (Invitrogen Life Technologies, Carlsbad, CA) and approximately 50 mg of frozen placental specimens on wet ice[79].

**Statistical approach**. Data were expressed as mean ± standard error of the mean (SEM). For the Z scores data were presented in min. to max. Statistical analyses were performed using Statistical Package for the Social Sciences (SPSS) software, Version 20.0 (SPSS Inc., Chicago, IL) and GraphPad Prism 6 (GraphPad Software, Inc., La Jolla, CA). Tests included repeated measures ANOVA, $t$ tests or one-way

ANOVA when relevant. Differences between the groups were assessed using Tukey's multiple comparisons post hocs. When appropriate, non-parametric tests such as Mann−Whitney and Kruskal−Wallis were used. Linear regression and $r$ values were determined in GraphPad Prism 6 (GraphPad Software, Inc., La Jolla, CA).

**Data availability**. The Agilent miR microarray data have been deposited with GEO-NCBI under the accession number GSE110597 and the Illumina RNA-seq data have been deposited in the Sequence Read Archive (SRA), using the NCBI portal, under the BioProject accession number PRJNA434509 and SRA accession number SRP133035.

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

## Acknowledgements

We thank Mr. Sharon Ovadia for his devoted assistance with animal care. We also thank Lisa Tietze for her help with in situ hybridization; Daniela Harbich, Andrea Parl, and Carola Eggert for their help with western blots/stainings; Michael Lüthi for help in human placental RNA isolation and qPCR measurements and Stoyo Karamihalev for help with the graphic images. The authors also wish to express their gratitude to the patients, physicians, and midwives from the Lindenhofgruppe, Bern, for participating in this study. This work was supported by: an FP7 Grant from the European Research Council (260463, A.C.); a research grant from the Israel Science Foundation (1565/15, A.C.); the ERANET Program, supported by the Chief Scientist Office of the Israeli Ministry of Health (3-11389, A.C.); the project was funded by the Federal Ministry of Education and Research under the funding code 01KU1501A (A.C.); research support from Roberto and Renata Ruhman (A.C.); research support from Bruno and Simone Licht; I-CORE Program of the Planning and Budgeting Committee and The Israel Science Foundation (grant no. 1916/12 to A.C.); the Nella and Leon Benoziyo Center for Neurological Diseases (A.C.); the Henry Chanoch Krenter Institute for Biomedical Imaging and Genomics (A.C.); the Perlman Family Foundation, founded by Louis L. and Anita M. Perlman (A.C.); the Adelis Foundation (A.C.); the Marc Besen and the Pratt Foundation and the Irving I. Moskowitz Foundation (A.C.). C.A. was supported by the Swiss National Science Foundation (SNSF) through the National Center of Competence in Research (NCCR) TransCure, University of Bern, Switzerland, and the Swiss National Science Foundation (Grant No. 310030_149958). M.J. was supported by a NARSAD young investigator grant.

## Author contributions

M.S. designed the experiments and prepared the manuscript. A.C. supervised the project. M.S. performed the embryo transfers. M.S. and T.P. performed the behavioral experiments, RNA, DNA and protein extractions. T.P. and Y.D. killed the animals. M.S. performed the in vitro experiments. M.J. constructed the libraries and performed NGS. S. R., M.J. and S.B.-D. performed the bioinformatic predictions and analysis. C.A. and J.Z. organized the collection of the human placenta samples and J.Z. and A.L. performed gene expression analysis.

## Additional information

**Competing interests:** The authors declare no competing interests.

