## [Peer Review File · Nature Communications]

Reviewers' comments:

Reviewer #1 (Remarks to the Author):

This manuscript describes how prenatal stress (PNS) may affect activity-based anorexia (ABA) in male and female mice. Females tend to demonstrate greater variability in this testing than males. However, exposure to PNS abolishes such differences and improves this behavioral domain in females with no effects noted in males, who are already more resistant. The investigators then go on to show that placental expression of miR-340 is likely mediating such effects with this miR targeting *Gr*, *Cry2*, and *H3f3b* transcripts and also this miR can affect nutrient transporter in mice and human placentae. The expression of miR-340 is likely mediated by DNA methylation, as the investigators show by several *in vivo* and *in vitro* procedures. Lastly, the investigators used placental-specific lentiviral transgenes and embryo transfer to upregulate miR-340, which led to increase susceptibility to ABA.

While the overall data on the potential effects of miR-340 are convincing, there is concern about the initial findings in that PNS actually protects females against ABA. It is difficult to interpret this seemingly paradoxical finding in light of several other studies, especially Bale's group who have shown that a similar PNS strategy actually leads to heightened anxiety with stimulation of the HPA axis, especially in males. The current findings are seemingly suggesting that PNS has beneficial effects in females. Before such a big conclusion can be drawn, the investigators need to do additional studies to examine other behavioral domains and metabolic pathways. At the least, anxiety testing and additional examination of the HPA axis should be performed to analyze how the current findings compare to well-established results with the same prenatal testing procedure done by Bale's group. The investigators should also test the pups in a metabolic cage and home cage activity to determine whether there are differences in metabolism and general activity that could account for potential differences. The other potential concern as it relates to these seemingly conflicting findings is whether the investigators documented intrauterine position, which could account for the initial variability observed in control females. Were these ones between two males (2M) *in utero*? If intrauterine position was not documented, this should be stated and addressed as a potential confounder.

The other major concern is the placental specific lentiviral system injection. In these studies, blastocysts stage embryos were incubated with lentivirus for miR-340 and then infected blastocysts were intrauterine transferred into pseudo-pregnant females. In Fig 5B, the investigators show a nice picture where the lentivirus overall appears to be confined to the discoid fetal placenta. However, Sadovsky has shown in the human that other trophoblast miRNAs (trophomiRs) can be packaged into exosomes or other vesicles, where they can cross into the maternal circulation. While this process appears to be specific to those miRNAs on primate-specific chromosome 19, it is possible, that overexpression of other miRNAs might result in similar processes, including in the mouse. Thus, the expression pattern of miR-340 should be examined in the dams to eliminate this potential confounder. It also seems from the Fig 5B that the lentiviral expression may be in the junctional zone and extending into the decidua, which could place it closer to the underlying uterine tissue.

Other Comments:

1. Might consider ending the Introduction with the hypotheses to be tested rather than reporting the actual results in the Introduction.
2. The terminology for genes/transcript expression in the mouse should be uppercase used for the first letter and lowercase for all other letters and then italicized.
3. The Discussion should present and discuss other findings when the same PNS paradigm model is used. Potential explanation for conflicting data should be provided.

4. In terms of the potential masculinized response observed in PNS females, this relates back to the comment above about the in utero position. Without any evidence to suggest that these females were actually masculinized, such as anogenital distance (AGD) or serum testosterone concentrations, such speculation should be downplayed.

5. Did the dams who were subjected to the PNS protocol demonstrate any evidence of anxiety-like behaviors or altered parental care that could have detrimentally influenced her pups?

6. For the statistical analyses, there is concern whether the data were controlled for potential litter effects. As the dam was subjected to the initial treatment in the form of PNS, the results of from all her pups should be nested under her in that the dam not the pup should be the experimental unit, which can be done using a split plot in same and time (for procedures involving multiple testing). It is also not clear in this regard how many male and female pups from the same litter were tested. Even if only one male and one female sibling were tested, their results still should be nested under the one dam. If the pups originated from only a few litters, this could potentially bias the results. For instance, one dam could have demonstrated an exaggerated response to the PNS that in turn affected her pups. To eliminate such a possibility, it needs to be clarified how many pups from each litter were screened and verification that all of the results were controlled for potential litter effects. Additionally, some details on the dam behavioral patterns and maternal care provided to the pups for the control and PNS groups are essential.

Reviewer #2 (Remarks to the Author):

This is a very interesting paper which reveals the trans-generational epigenetic mechanisms for stress induced anorexia in an experimental mouse model. In this study the pregnant mother is subject to self-induced stress by activity in a running wheel. The anorexic effect is not induced in males sharing the same uterus, but is produced in 40% of females. In those females which do exhibit anorexia, maternal cortisol levels were found to be significantly higher than in males, and most notably, also higher in those females which were not protected from becoming anorexic. In this study the authors have further investigated the placental expressed genes that influence the future functioning of the next generation and its vulnerability to activity based anorexia (ABA). The non-coding RNA, Mir340, was found to undergo hyper-methylation in the female foetal placenta when the mother was stressed, and this in turn resulted in a predisposition for daughters to also develop stress induced anorexia.

The authors refer to the placenta as a "sexless" organ, but it should be noted that the genetic determination of the placenta is activated by Cdx2, a maternally expressed gene that is switched on in the fertilised egg to induce trophoblast development. This developmental initiation of placental development occurs before any of the spermatozoan genes are activated. Certainly, in its earliest stages of development the placenta is more epigenetic and genetically related to mother. This is a very interesting paper focussing the genetic origins for female anorexia arising in the pregnant mother. However, it leaves one question unanswered, namely "what is changed in the developing brain and differs between the two groups of mice when prenatally stressing the mother"? Cortisol is not a sex dependent hormone. However, there are a large number of imprinted genes which are either maternally, and others paternally, expressed in the placenta and expressed in the developing brain. Indeed, a number of these imprinted genes are developmentally co-expressed in the developing placenta and in the developing brain. Could these contribute to the code for explaining the sex bias for activity dependent anorexia?

Response to Reviewers

Reviewer #1 (Remarks to the Author):

“This manuscript describes how prenatal stress (PNS) may affect activity-based anorexia (ABA) in male and female mice. Females tend to demonstrate greater variability in this testing than males. However, exposure to PNS abolishes such differences and improves this behavioral domain in females with no effects noted in males, who are already more resistant. The investigators then go on to show that placental expression of miR-340 is likely mediating such effects with this miR targeting Gr, Cry2, and H3f3b transcripts and also this miR can affect nutrient transporter in mice and human placentae. The expression of miR-340 is likely mediated by DNA methylation, as the investigators show by several in vivo and in vitro procedures. Lastly, the investigators used placental-specific lentiviral transgenes and embryo transfer to upregulate miR-340, which led to increase susceptibility to ABA. While the overall data on the potential effects of miR-340 are convincing, there is concern about the initial findings in that PNS actually protects females against ABA. It is difficult to interpret this seemingly paradoxical finding in light of several other studies, especially Bale’s group who have shown that a similar PNS strategy actually leads to heightened anxiety with stimulation of the HPA axis, especially in males. The current findings are seemingly suggesting that PNS has beneficial effects in females. Before such a big conclusion can be drawn, the investigators need to do additional studies to examine other behavioral domains and metabolic pathways. At the least, anxiety testing and additional examination of the HPA axis should be performed to analyze how the current findings compare to well-established results with the same prenatal testing procedure done by Bale’s group. The investigators should also test the pups in a metabolic cage and home cage activity to determine whether there are differences in metabolism and general activity that could account for potential differences.”

We thank the reviewer for this analysis of our study. We would like to start by stating that by no means did we suggest that PNS has a positive effect on females. Although Bale’s group used a different developmental window to produce the most robust results in males, our findings are very much in line with theirs. Thus, our results are not contradictory to their (nor others’) findings, as we will try to elucidate below.

The observation that PNS females became resistant to the ABA protocol was at first unexpected, but after exploring the issue further, it became evident that this resistance may result from compensations that would help them survive in a food deprived environment at the high price of leaving them more susceptible to other diseases (for an example the metabolic abnormalities shown below). As suggested by Reviewer 1, we examined maternal and fetal corticosterone (CORT), gene expression in the fetal CRF system and maternal behavior. We found higher levels of CORT in female fetuses (Figure 2G in the manuscript) and in PNS dams (New Figure 2C in the manuscript, and Figure 1 below). Indeed, the fetal CRF system was dysregulated; as expected since it is a well-known sign of prenatal stress. In contrast, we did not observe any postnatal anxiety in the PNS dams as reflected by normal maternal behavior (7 observations spread over 3 hours, every 30 minutes in the dark cycle on postpartum weeks (PPW) 1 and 3) (Figure 1, below).

Figure 1

Next, because of our assumption that the observed resistance to ABA may be a “beneficial” side effect of a very negative overall outcome, we examined general activity and metabolism and challenged these females with different protocols (other than ABA). While the results of these experiments are beyond the scope of the present study, we can confirm that these females did not “benefit” from the PNS procedure. While they coped better than controls with food restriction and the ABA protocol, they were susceptible to metabolic disease, glucose intolerance and showed a chronic inflammatory state. For example, at 2 months of age and without any further challenge, PNS female offspring displayed higher BW, food consumption and adiposity, but normal metabolic rate. In addition, they were hyperactive, which contrary to our observations, should have in theory increased their vulnerability to ABA (Figure 2, below).

Figure 2

Thus, it appears that these PNS females are programmed to become overweight, rather than underweight, and this may be at least one of the reasons underlying their resistance to ABA.

Stress-related programming is a highly complex system and therefore it is imperative to address the distinction between different types of early life stress and how they may affect subsequent predisposition to eating disorders. For example, it has been suggested that trauma in childhood may serve as a trigger for an existing predisposition to certain EDs, whereas gestational or pre-gestational trauma may relate more to programming the predisposition or protecting from it, depending on the severity, timing and extent of the insult¹. This complexity was also raised in a recent study that observed an increased risk of bulimia nervosa and mixed eating disorders among girls who were exposed to either prenatal or early postnatal life stress. However, this was not true for anorexia nervosa². Thus, our results are not necessarily paradoxical, given that early life stress

taking place at different developmental windows and with different levels of severity may exert potentially very different effects on the susceptibility for the different eating disorders in the offspring.

Since this study was designed to identify a placental candidate mediating the predisposition to ABA, we had not included these data in the manuscript. However, we can happily add it if it is felt necessary.

“The other potential concern as it relates to these seemingly conflicting findings is whether the investigators documented intrauterine position, which could account for the initial variability observed in control females. Were these ones between two males (2M) in utero? If intrauterine position was not documented, this should be stated and addressed as a potential confounder.”

This is an excellent point. We indeed documented intrauterine position, and in fact chose to only analyze females located in the center of the uterine horn and surrounded by a male and a female to minimize variability. When analyzing miR-340 according to the sex of the adjacent fetus, we found that females located between 2 males presented the lowest levels of miR-340, suggesting prenatal androgen exposure may influence placental miR-340 levels. We added this information in the methods section and provided a new figure (New Figure 3N, in the manuscript).

“The other major concern is the placental specific lentiviral system injection. In these studies, blastocysts stage embryos were incubated with lentivirus for miR-340 and then infected blastocysts were intrauterine transferred into pseudo-pregnant females. In Fig 5B, the investigators show a nice picture where the lentivirus overall appears to be confined to the discoid fetal placenta. However, Sadovsky has shown in the human that other trophoblast miRNAs (trophomiRs) can be packaged into exosomes or other vesicles, where they can cross into the maternal circulation. While this process appears to be specific to those miRNAs on primate-specific chromosome 19, it is possible, that overexpression of other miRNAs might result in similar processes, including in the mouse. Thus, the expression pattern of miR-340 should be examined in the dams to eliminate this potential confounder. It also seems from the Fig 5B that the lentiviral expression may be in the junctional zone and extending into the decidua, which could place it closer to the underlying uterine tissue.”

We thank the reviewer for this comment. Since the process of vesicular transfer appears to be specific for primate specific chromosome 19 miRNAs, as mentioned by the reviewer, this appears to be a rather unlikely possibility in the mouse. In addition, from the many placentas that we examined in preliminary studies to reach the appropriate viral concentration for a homogenous infection, we observed that GFP is not ubiquitously expressed (see Figure S5). In the left picture, it can be seen that the infection is weak in the center of the top layer (maternal decidua). This is also the case for the labyrinth (right picture), where GFP is not observed in the central area. Thus, miR-340 OE is concentrated in the junctional area of the placenta; any minimal infection seen in the decidua or in the uterus could be the direct result of the medium that may contain leftover virus. In any case, in the unlikely event that miR-340 may have been transferred from the placenta into the mothers' circulation; we have no reason to assume this would have any direct impact on the phenotype of the fetuses.

However, in order to explore the possibility that miR-340 may undergo vesicular transfer, we examined miR-340 levels in the fetal brain. We focused on the hypothalamus of the groups displaying high and low endogenous expression of placental miR-340 and in the placental transgenes with miR-340 OE. We found no differences in miR-340 levels between the groups (Figure 3, below), suggesting miR-340 is not secreted by the placenta to reach the fetal brain. We have not included these data in the manuscript due to space limitations, but again, they can be added upon request.

Figure 3

Other Comments:

1. "Might consider ending the Introduction with the hypotheses to be tested rather than reporting the actual results in the Introduction."

Thank you for this comment; we have now clearly stated our hypothesis, but left a summary of our findings in the end of the introduction.

2. "The terminology for genes/transcript expression in the mouse should be uppercase used for the first letter and lowercase for all other letters and then italicized."

Thank you, we have corrected this mistake throughout the text.

3. "The Discussion should present and discuss other findings when the same PNS paradigm model is used. Potential explanation for conflicting data should be provided."

We attempted to cite as many stress protocols as we found in the context of eating disorders. It is difficult to find exactly the same protocol used by our group in this context given the relatively limited amount of studies in the field of EDs and PNS. We intended to highlight how differences in the protocols and in the windows of intervention may lead to different outcomes. Regarding potentially conflicting data, we believe this is the first study that shows that predisposition to anorexia originates in utero through placental programming. We found one study that showed that, contrary to ours, prenatal stress increases susceptibility to ABA, but only in a subset of passively coping rats and without providing fetal or placental mechanistic insights⁵. However, that study used late gestation stress, which differs significantly from our protocol. In fact, we similarly found in our previous study that stress during late gestation increases susceptibility to other ED (binge eating)⁶ further emphasizing the importance of the type and timing of the stressor in inducing or preventing susceptibility to EDs. We have added this discussion to the conclusion.

4. "In terms of the potential masculinized response observed in PNS females, this relates back to the comment above about the in utero position. Without any evidence to suggest that these females were actually masculinized, such as anogenital distance (AGD) or serum testosterone concentrations, such speculation should be downplayed."

In the case of the PNS females, levels of miR-340 were low regardless of the sex of the adjacent fetus, as can be seen by the low variation in the group (Figure 2J). Regarding the anogenital distance, we were unable to accurately measure it, but we did document that males were indistinguishable from females in the PNS group at birth due to larger than usual anogenital distance in the female newborns (we had no difficulty distinguishing males from females in the control group). We were able to confirm the sex of the PNS offspring a week later. We have added a comment about this in the manuscript.

5. "Did the dams who were subjected to the PNS protocol demonstrate any evidence of anxiety-like behaviors or altered parental care that could have detrimentally influenced her pups?"

Dams subjected to our PNS protocol were unmistakably stressed, easily startled and presented high levels of CORT compared to controls. We did not perform anxiety tests during gestation (because it would have interfered with the protocol) or during lactation (so as not to disturb the offspring). However, since we did not detect abnormalities in maternal care in this group (please refer to our answer to comment 1, Figure 1 above) we disregarded the possibility of postnatal detrimental care influences on the offspring.

6. "For the statistical analyses, there is concern whether the data were controlled for potential litter effects. As the dam was subjected to the initial treatment in the form of PNS, the results of from all her pups should be nested under her in that the dam not the pup should be the experimental unit, which can be done using a split plot in same and time (for procedures involving multiple testing). It is also not clear in this regard how many male and female pups from the same litter were tested. Even if only one male and one female sibling were tested, their results still should be nested under the one dam. If the pups originated from only a few litters, this could potentially bias the results. For instance, one dam could have demonstrated an exaggerated response to the PNS that in turn affected her pups. To eliminate such a possibility, it needs to be clarified how many pups from each litter were screened and verification that all of the results were controlled for potential litter effects.

Additionally, some details on the dam behavioral patterns and maternal care provided to the pups for the control and PNS groups are essential.”

Thank you for this comment and we apologize for unintentionally omitting this information. We used animals selected by their uterine position and the sex of the adjacent fetus for our analysis. We only analyzed more than one fetus per litter when we intended to compare intra-litter effects and explore endogenous variability; otherwise, only 1 fetus per litter was analyzed (from the central uterine location and surrounded by a male and a female); considering that we were also interested in within-litter effects, averaging each litter in these cases would have been counterproductive. In the case of the adult animals, all litters were split between the different groups (unchallenged, food restricted, and ABA protocol) in a randomized complete block design to avoid potential litter effects. We have now added a detailed explanation in the methods section.

Reviewer #2 (Remarks to the Author):

“This is a very interesting paper which reveals the trans-generational epigenetic mechanisms for stress induced anorexia in an experimental mouse model. In this study the pregnant mother is subject to self-induced stress by activity in a running wheel. The anorexic effect is not induced in males sharing the same uterus, but is produced in 40% of females. In those females which do exhibit anorexia, maternal cortisol levels were found to be significantly higher than in males, and most notably, also higher in those females which were not protected from becoming anorexic. In this study the authors have further investigated the placental expressed genes that influence the future functioning of the next generation and its vulnerability to activity based anorexia (ABA). The non-coding RNA, Mir340, was found to undergo hyper-methylation in the female foetal placenta when the mother was stressed, and this in turn resulted in a predisposition for daughters to also develop stress induced anorexia. The authors refer to the placenta as a "sexless" organ, but it should be noted that the genetic determination of the placenta is activated by Cdx2, a maternally expressed gene that is switched on in the fertilised egg to induce trophoblast development. This developmental initiation of placental development occurs before any of the spermatozoan genes are activated. Certainly, in its earliest stages of development the placenta is more epigenetic and genetically related to mother.

This is a very interesting paper focussing the genetic origins for female anorexia arising in the pregnant mother.”

We thank the reviewer for acknowledging the importance of our manuscript.

“However, it leaves one question unanswered, namely "what is changed in the developing brain and differs between the two groups of mice when prenatally stressing the mother"? Cortisol is not a sex dependent hormone. However, there are a large number of imprinted genes which are either maternally, and others paternally, expressed in the placenta and expressed in the developing brain. Indeed, a number of these imprinted genes are developmentally co-expressed in the developing placenta and in the developing brain. Could these contribute to the code for explaining the sex bias for activity dependent anorexia?”

Thank you for this comment. The question of what changes in the developing brain in response to prenatal stress is of high interest to us; it is extremely complex and is therefore the focus of current work being performed in the lab on a variety of different projects. While exploring the role played by imprinted genes in this phenotype is beyond the scope of the present study, we explored how the natural variability in the maternal environment leading to different levels of placental miR-340 affects the fetal brain leading to the observed predisposition. In order to examine this, we focused on the hypothalamus (the main brain area involved in appetite regulation and energy balance) and on genes specifically linked to anorexia in women and in different animal models of anorexia as possible candidates. Hypothalamic dysregulation in anorexia has been reported for the melanocortin, the neurotrophic and the serotonin systems⁷⁻¹² and the HPA axis¹³⁻¹⁵. We chose representative genes of those systems that repeatedly emerged as potential mediators of the anorexic phenotype:

- 1) Agouti-related peptide (AgRP)^{8,16-18},*
- 2) Melanocortin receptor 4 (Mc4R)¹⁹,*
- 3) Serotonin receptor (Htr1a)^{12,20-22},*
- 4) Brain-derived neurotrophic factor (Bdnf)^{10,23,24},*
- 5) Arginine vasopressin (Avp)^{14,15}*
- 6) Hypocretin/Orexin (Hcrt)^{5,25,26}.*

*When exploring the expression of these candidate genes in the PNS group, we found abnormal gene expression in the hypothalamus of the fetuses, with dramatic increases in 4 out of the 6 candidate genes ($F_{(6,11)}=33.95$, $p<0.000$). However, expression changes did not carry on through adulthood and accordingly the ABA phenotype did not differ from control resistant females beyond *Bdnf* (known to be downregulated by PNS²⁷) (Figure 4, below). Further exploration of this group revealed long lasting inflammation of the hypothalamus that may have led these females to be resistant to anorexia but prone to other diseases (see for example Figure 1 in the response to the first comment to Reviewer 1). Since the focus of the current project was to detect a placental candidate gene and not to focus on the PNS manipulation, we believe these data may distract the reader from the main story and we therefore chose not to include it in the manuscript. The complexity of the PNS group led to a further project centred on this question, which will be discussed in future publications. However, these data can be added upon request.*

Figure 4

Since the main focus on this project was examining the variability in the control group, we explored the global expression of the AN candidate genes in fetuses with high and low placental miR-340, under undisturbed and manipulated conditions (lenti). We found that the expression of these genes differed according to placental miR-340 levels ($F_{(6,5)}=13.1$, $p=0.006$). Specifically, we found that hypothalamic AgRP and Hcrt were highly expressed in control/untreated fetuses with high levels of miR-340 compared to fetuses with low levels of miR-340. In contrast, Htr1a and Bdnf were decreased (New Figure 3R, in the manuscript). Remarkably, miR-340 OE transgene fetuses showed a similar pattern in particular for AgRP and Htr1a ($F_{(6,5)}=7.08$, $p=0.024$) (New figure 5C, in the manuscript), suggesting that abnormal expression of these genes during brain development as a result of placental miR-340 OE may be involved in later life predisposition to anorexia.

To further explore the long term effects of different levels of placental miR-340 and fetal hypothalamic abnormalities in the expression of candidate anorexia genes in animals with and without ABA, we analysed the expression of these genes in the adult females. ABA females presented abnormal gene expression ($F_{(6,5)}=24.60$, $p=0.001$), with reduced levels of AgRP and increased levels of Htr1a. This opposite pattern of expression when compared to the fetuses may represent a form of long term overcompensation. Bdnf levels remained low and Hcrt levels were drastically upregulated in this group, similarly to fetuses with high levels of placental miR-340 (New figure 1J, in the manuscript). Interestingly, gene expression also differed between resistant and ABA transgenes ($F_{(6,5)}=53.83$, $p<0.000$). However, only the pattern of expression of AgRP and Htr1a were reproduced. Thus, our results confirm the central role played by the hypothalamic

melanocortin and serotonin systems in anorexia, which may be mediated by placental miR-340 (New figure 6I, in the manuscript).

References

1. St-Hilaire, A. *et al.* A prospective study of effects of prenatal maternal stress on later eating-disorder manifestations in affected offspring: preliminary indications based on the Project Ice Storm cohort. *Int. J. Eat. Disord.* **48**, 512–6 (2015).
2. Su, X. *et al.* Prenatal and early life stress and risk of eating disorders in adolescent girls and young women. *Eur Child Adolesc Psychiatry* **25**, 1245–1253 (2016).
3. Rojo, L., Conesa, L., Bermudez, O. & Livianos, L. Influence of Stress in the Onset of Eating Disorders: Data From a Two-Stage Epidemiologic Controlled Study. *Psychosom. Med.* **68**, 628–635 (2006).
4. Caslini, M. *et al.* Disentangling the Association Between Child Abuse and Eating Disorders. *Psychosom. Med.* **78**, 79–90 (2016).
5. Boersma, G. J. *et al.* Failure to upregulate *Agrp* and *Orexin* in response to activity based anorexia in weight loss vulnerable rats characterized by passive stress coping and prenatal stress experience. *Psychoneuroendocrinology* **67**, 171–181 (2016).
6. Schroeder, M. *et al.* A Methyl-Balanced Diet Prevents CRF-Induced Prenatal Stress-Triggered Predisposition to Binge Eating-like Phenotype. *Cell Metab.* **25**, 1269–1281.e6 (2017).
7. Johansen, J. E. *et al.* Evidence for hypothalamic dysregulation in mouse models of anorexia as well as in humans. *Physiol. Behav.* **92**, 278–282 (2007).
8. Adan, R. A. H. *et al.* Melanocortin system and eating disorders. *Ann. N. Y. Acad. Sci.* **994**, 267–74 (2003).
9. Gratacòs, M. *et al.* Role of the neurotrophin network in eating disorders' subphenotypes: Body mass index and age at onset of the disease. *J. Psychiatr. Res.* **44**, 834–840 (2010).
10. Mercader, J. M. *et al.* Blood levels of brain-derived neurotrophic factor correlate with several psychopathological symptoms in anorexia nervosa patients. *Neuropsychobiology* **56**, 185–90 (2007).
11. Kaye, W. Neurobiology of anorexia and bulimia nervosa. *Physiol. Behav.* **94**, 121–35 (2008).

12. Bailer, U. F. *et al.* Altered brain serotonin 5-HT_{1A} receptor binding after recovery from anorexia nervosa measured by positron emission tomography and [carbonyl¹¹C]WAY-100635. *Arch. Gen. Psychiatry* **62**, 1032–41 (2005).
13. Lawson, E. A. *et al.* Increased hypothalamic-pituitary-adrenal drive is associated with decreased appetite and hypoactivation of food-motivation neurocircuitry in anorexia nervosa. *Eur. J. Endocrinol.* **169**, 639–47 (2013).
14. Pei, H., Sutton, A. K., Burnett, K. H., Fuller, P. M. & Olson, D. P. AVP neurons in the paraventricular nucleus of the hypothalamus regulate feeding. *Mol. Metab.* **3**, 209–15 (2014).
15. Gold, P. W., Kaye, W., Robertson, G. L. & Ebert, M. Abnormalities in plasma and cerebrospinal-fluid arginine vasopressin in patients with anorexia nervosa. *N. Engl. J. Med.* **308**, 1117–23 (1983).
16. Vink, T. *et al.* Association between an agouti-related protein gene polymorphism and anorexia nervosa. *Mol. Psychiatry* **6**, 325–8 (2001).
17. Yilmaz, Z. *et al.* The role of leptin, melanocortin, and neurotrophin system genes on body weight in anorexia nervosa and bulimia nervosa. *J. Psychiatr. Res.* **55**, 77–86 (2014).
18. Nilsson, I. A. K. *et al.* Evidence of hypothalamic degeneration in the anorectic anx/anx mouse. *Glia* **59**, 45–57 (2011).
19. Gutiérrez, E. *et al.* High ambient temperature reverses hypothalamic MC4 receptor overexpression in an animal model of anorexia nervosa. *Psychoneuroendocrinology* **34**, 420–9 (2009).
20. Kaye, W. H. *et al.* Serotonin alterations in anorexia and bulimia nervosa: New insights from imaging studies. *Physiol. Behav.* (2005). doi:10.1016/j.physbeh.2005.04.013
21. Galusca, B. *et al.* Organic background of restrictive-type anorexia nervosa suggested by increased serotonin 1A receptor binding in right frontotemporal cortex of both lean and recovered patients: [18F]MPPF PET scan study. *Biol. Psychiatry* **64**, 1009–13 (2008).
22. Haleem, D. J. Serotonin neurotransmission in anorexia nervosa. *Behav. Pharmacol.* **23**, 478–95 (2012).
23. Brandys, M. K., Kas, M. J. H., van Elburg, A. A., Campbell, I. C. & Adan, R. A. H. A meta-analysis of circulating BDNF concentrations in anorexia nervosa. *World J. Biol. Psychiatry* **12**, 444–54 (2011).
24. Ribasés, M. *et al.* Association of BDNF with anorexia, bulimia and age of onset of weight loss in six European populations. *Hum. Mol. Genet.* **13**, 1205–12 (2004).
25. Bronsky, J. *et al.* Changes of orexin A plasma levels in girls with anorexia nervosa during eight weeks of realimentation. *Int. J. Eat. Disord.* **44**, 547–52 (2011).
26. Janas-Kozik, M. *et al.* Plasma levels of leptin and orexin A in the restrictive type of anorexia nervosa. *Regul. Pept.* **168**, 5–9 (2011).
27. Boersma, G. J. *et al.* Prenatal stress decreases Bdnf expression and increases methylation of Bdnf exon IV in rats. *Epigenetics* **9**, 437–47 (2014).

REVIEWERS' COMMENTS:

Reviewer #1 (Remarks to the Author):

The revised manuscript is improved relative to the previous version. The investigators have been highly responsive to the Reviewers' comments. A few issues though remain in light of the additional data presented in Figures 1 and 2 in the response to the Reviewers.

The data presented in these Figures should be included as supplementary material for the manuscript and discussed within the Discussion. Based on these additional data, some of the concluding statements regarding PNS effects on females should be toned down and these additional results considered in the final Concluding statements.

While Figure 3 somewhat addresses the potential vesicular transfer within the conceptus, it still does not address it though from the fetal to maternal placenta transfer perspective, which is the bigger concern in terms of effects on the dam. At the least, this potential confounder should be addressed in the Discussion.

In regards to the anogenital distance (AGD), it is not clear what to make of this seemingly transient finding. Has this been noted by others?

While it is appreciated that the authors have added additional references and comparisons to other studies, the Discussion should provide a better holistic context for these studies relative to other well-documented maternal rodent stress studies, in particular from the Bales' group.

For the statistical analyses, the updated sentences suggest that no male and female siblings were tested, which seems surprising. How were the F1 males selected in terms of their intrauterine location and position relative to female siblings? If even one male and one female sibling were tested, these data should be nested together under the single dam.

Reviewer #2 (Remarks to the Author):

This remains a very interesting paper and this has been demonstrated by the enthusiastic response of the referees. Of course the findings raise many future questions to be addressed, but it is clear from the authors response to such questions that they have them in mind and that further studies are progressing in this direction.

I was not suggesting that genomic imprinting might provide the answers, but to also have such methylation silencing in mind, particularly when considering the placenta. Many X-linked genes are expressed in the placenta, and sons always inherit their single X chromosome from mother. Moreover, many of the paternal X-linked genes which are expressed in the placenta are also expressed in the developing brain, most notably the hypothalamus, some of which may fail to escape from demethylation reprogramming during the early post fertilisation period.

I had no problems with this paper. The authors are investigating a very interesting and complex subject, and their studies have addressed part of this complexity, namely the predisposition of males to exhibit anorexia nervosa. I strongly recommend publication.

Response to the Reviewers' comments:

Reviewer #1:

The revised manuscript is improved relative to the previous version. The investigators have been highly responsive to the Reviewers' comments. A few issues though remain in light of the additional data presented in Figures 1 and 2 in the response to the Reviewers.

The data presented in these Figures should be included as supplementary material for the manuscript and discussed within the Discussion. Based on these additional data, some of the concluding statements regarding PNS effects on females should be toned down and these additional results considered in the final Concluding statements.

We thank the Reviewer for appreciating our considerable efforts to improve the manuscript. We have added these data in Supplementary Fig. 1 and described them in the results section on page 6.

While Figure 3 somewhat addresses the potential vesicular transfer within the conceptus, it still does not address it though from the fetal to maternal placenta transfer perspective, which is the bigger concern in terms of effects on the dam. At the least, this potential confounder should be addressed in the Discussion.

We have added the following to the methods section: "This technique is highly specific to the transduction of the junction, therefore making vesicular transfer highly unlikely". We prefer this alternative solution as we feel it is sufficient to answer the above point yet avoids adding to an already complicated discussion.

In regards to the anogenital distance (AGD), it is not clear what to make of this seemingly transient finding. Has this been noted by others?

We did not mean that the effect is transient, but that the difficulty in identifying the sex of the newborn is transitory. On the 2nd postnatal week, the clear appearance of the nipples in the female offspring allowed us for clear identification. In utero programmed anogenital distance is actually known to be maintained during postnatal development (Bánszegi et al 2009; 2010, Hotchkiss et al 2007). We have rephrased the sentence to make this clearer.

While it is appreciated that the authors have added additional references and comparisons to other studies, the Discussion should provide a better holistic context for these studies relative to other well-documented maternal rodent stress studies, in particular from the Bales' group.

It is undeniable that discussing the differences between stress models and how they may affect eating disorder susceptibility would be very interesting. We have cited as much relevant work as possible in the context of early life stress and anorexia nervosa. However, due to space limitations and focus, we are unable to add a further discussion about prenatal stress models. Besides, we think it is beyond the scope of the present manuscript and may distract the reader away from our

main focus. In future studies, we aim to address the interesting topic of prenatal stress and metabolism.

For the statistical analyses, the updated sentences suggest that no male and female siblings were tested, which seems surprising. How were the F1 males selected in terms of their intrauterine location and position relative to female siblings? If even one male and one female sibling were tested, these data should be nested together under the single dam.

We apologize that our statement was unclear. When we wrote that only 1 fetus per litter was analyzed, we meant 1 for each sex. We have updated this section again, explaining that the chosen males had 1 male and 1 female as adjacent fetuses exactly mirroring the selection criterion for the chosen females. Regarding averaging males with females for each litter, we feel this would be counterproductive considering our interest in sex differences.

Reviewer #2:

This remains a very interesting paper and this has been demonstrated by the enthusiastic response of the referees. Of course the findings raise many future questions to be addressed, but it is clear from the authors response to such questions that they have them in mind and that further studies are progressing in this direction.

I was not suggesting that genomic imprinting might provide the answers, but to also have such methylation silencing in mind, particularly when considering the placenta. Many X-linked genes are expressed in the placenta, and sons always inherit their single X chromosome from mother. Moreover, many of the paternal X-linked genes which are expressed in the placenta are also expressed in the developing brain, most notably the hypothalamus, some of which may fail to escape from demethylation reprogramming during the early post fertilisation period. I had no problems with this paper. The authors are investigating a very interesting and complex subject, and their studies have addressed part of this complexity, namely the predisposition of males to exhibit anorexia nervosa. I strongly recommend publication.

We thank the Reviewer for acknowledging the importance and complexity of our work and for providing interesting advice, which we will look into in our future studies.